# Social complexity, life-history and lineage influence the molecular basis of castes in vespid wasps

Christopher Douglas Robert Wyatt [1] ✉, Michael Andrew Bentley[1,5], Daisy Taylor[2,5], Emeline Favreau [1], Ryan Edward Brock [2,3], Benjamin Aaron Taylor[1], Emily Bell [2], Ellouise Leadbeater [4] & Seirian Sumner [1] ✉

A key mechanistic hypothesis for the evolution of division of labour in social insects is that a shared set of genes co-opted from a common solitary ancestral ground plan (a genetic toolkit for sociality) regulates caste differentiation across levels of social complexity. Using brain transcriptome data from nine species of vespid wasps, we test for overlap in differentially expressed caste genes and use machine learning models to predict castes using different gene sets. We find evidence of a shared genetic toolkit across species representing different levels of social complexity. We also find evidence of additional fine-scale differences in predictive gene sets, functional enrichment and rates of gene evolution that are related to level of social complexity, lineage and of colony founding. These results suggest that the concept of a shared genetic toolkit for sociality may be too simplistic to fully describe the process of the major transition to sociality.

The major evolutionary transitions span all levels of biological organisation, facilitating the evolution of life's complexity on earth via cooperation between single entities (e.g. genes in a genome, cells in a multicellular body, insects in a colony), generating fitness benefits beyond those attainable by a comparable number of isolated individuals[1]. The evolution of eusociality is one of the major transitions and is of general relevance across many levels of biological organisation from genes assembled into genomes, single cells into multicellular entities, and insects cooperating in superorganismal societies. The best-studied examples of sociality are in the hymenopteran insects (bees, wasps and ants) - a group of over 17,000 species, exhibiting different forms of sociality across the transition from simple sociality (with small societies where all group members are able to reproduce and switch roles in response to opportunity), through to complex societies (consisting of thousands of individuals, each committed during development to a specific cooperative role and working for a shared reproductive outcome within the higher-level 'individual' of the colony, known as the 'superorganism'[2]). Recent analyses of the molecular mechanisms of insect sociality have revealed how conserved suites of genes, networks and functions are shared among independent evolutionary events of insect superorganismality[3–8]. An outstanding question is whether there are fine-scale differences in the genetic mechanisms operating at different levels of complexity across the spectrum of sociality – from the simplest to most complex societies.

A key step in the evolution of sociality is the emergence of a reproductive division of labour, where some individuals commit to reproductive or non-reproductive roles, known as queens and workers respectively in the case of insect societies. An overarching mechanistic hypothesis for social evolution is that the repertoire of behaviours typically exhibited in the life cycle of the solitary ancestor of social species were uncoupled to produce a division of labour among group

[1]Centre for Biodiversity and Environment Research, Dept Genetics, Evolution & Environment, University College London, London WC1E 6BT, UK. [2]School of Biological Sciences, University of Bristol, Bristol BS8 1TQ, UK. [3]Department of Crop Genetics, John Innes Centre, Norwich Research Park, Norwich, Norfolk NR4 7UH, UK. [4]Department of Biological Sciences, Royal Holloway University of London, Egham TW20 0EX, UK. [5]These authors contributed equally: Michael Andrew Bentley, Daisy Taylor. ✉e-mail: c.wyatt@ucl.ac.uk; s.sumner@ucl.ac.uk

members with individuals specialising in either the reproductive ('queen') or provisioning ('worker') phases of the solitary ancestor[9]. Such phenotypic decoupling implies that there will be a conserved mechanistic toolkit that regulates queen and worker phenotypes in species representing different levels of social complexity across the spectrum of the major transition (reviewed in[10]). An alternative to the shared toolkit hypothesis is that the molecular processes regulating social behaviours in non-superorganismal societies (where caste remains flexible, and selection acts primarily on individuals) differ fundamentally from those processes that regulate social behaviours in superorganismal societies[11,12]. Phenotypic innovations across the animal kingdom have been linked to genomic evolution: taxonomically-restricted genes[13–17], rapid evolution of proteins[18,19] and regulatory elements[18,20] have been found in most lineages of social insects[21]. Indeed, some recent studies have suggested that the processes regulating different levels of social complexity may be different[8,18,20,22]. The innovations in social complexity and the shift in the unit of selection (from individual- to group-level[23]) that occur in a major transition may therefore be accompanied by shifts in genomic processes and evolution, throwing into question whether the concept of a universal conserved genomic toolkit that regulates social behaviours across the spectrum of the major transition may be too simplistic[24,25]. The roles of conserved and novel processes are not necessarily mutually exclusive; novel processes may coincide with phenotypic innovations, whilst conserved mechanisms may regulate core processes at all stages of social evolution.

Until recently, the data available to test these hypotheses were largely limited to species that represent either the most complex – superorganismal - forms of sociality (e.g. most ants, honeybees[26]), or the simplest forms of social complexity as non-superorganisms (e.g. *Polistes* wasps[5,27–29] and incipiently social bees[30–33]), the latter of which may represent the first stages in the major transition. Recent studies have identified core gene sets underlying caste-differentiated brain gene expression across a range of ants[6] and bees[8]; ants lack ancestrally non-superorganismal representatives and so may not be informative for studying simpler societies and, putatively, the process of social evolution[34]. Analyses of molecular evolution across two independently evolved lineages of sociality in bees showed increased levels of regulatory complexity with social complexity[8]. These insights highlight the need for a fine-scale dissection of the molecular processes regulating social behaviour for species representing different forms of social complexity[8], but also the need to consider the influence of life-history traits on gene evolution, as a selective process that is independent of complexity per se.

A promising group for unpicking the molecular basis of social complexity are the vespid wasps[35], which exhibit the full diversity of social complexity among some 1,100 species, as well as diversity in life-history traits such as mode of colony founding. Aculeate wasps are of paramount importance in studying the evolution of sociality in the Hymenoptera as they represent the evolutionary root to both bees and ants[36]. However, to date sociogenomic studies of caste evolution in wasps have been limited to *Polistes*[5,14,28], thus not yet permitting a test of the toolkit hypothesis beyond this lineage[5].

Here we generate brain transcriptomic data of caste-specific phenotypes for nine species of social wasps, representing diversity in social complexity and mode of colony founding, a life-history trait that is independent of social complexity (Fig. 1). Brain tissue is appropriate here, as it is the epicentre of behavioural regulation and is known to be highly dynamic and responsive to social interactions and social behaviour[37]; moreover, behaviour is one of the defining features of castes and brain-associated gene expression studies in social insects[10]. We employ a machine-learning algorithm - support vector machines (SVMs) – along-side more conventional gene expression analyses to conduct fine-scale analysis of the molecular processes associated with caste, to test the hypothesis that there is a conserved genetic toolkit for social behaviour in species displaying different forms of sociality among putative proxy representatives of non-superorganismal (simple) to superorganismal (complex) species in vespid wasps (Aim1; Hypothesis 1). In Aim 2 we further interrogate these data to explain shared and contrasting patterns of transcription among species. Specifically, we test the hypothesis that transcriptomic signatures of social behaviour may be influenced by level of social complexity (i.e. simpler versus the more complex societies) (Hypothesis 2;[24]); and further test the possible effects of lineage (vespines versus polistines) (Hypothesis 3) and life-history on gene expression, specifically with respect to the mode of colony foundation (i.e. independent-founding versus swarm-founding) (Hypothesis 4). Finally, we assess whether the levels of social complexity are associated with contrasting rates of gene evolution[24] and if worker-biased genes are undergoing more rapid evolution, as reported in bees[8,13,17,19,20] and ants[6,7] (Hypothesis 5). Our results provide evidence of a conserved genetic toolkit across forms of social complexity in the spectrum of the major transition to sociality in wasps; they also reveal how there are fine-scale differences in the molecular patterns and processes between simple and complex societies[24], and with a life-history trait. These results suggest the concept of a shared genetic toolkit across the spectrum of sociality and across lineages may be too simplistic, and that level of social complexity, life-history traits and lineage influence the molecular processes underpinning social behaviour.

## Results

We chose one species from each of nine different genera of social wasps from among the two main subfamilies - Polistinae and Vespinae; collectively these species represent the full spectrum of social diversity (detail on species choice given in Fig. 1 and Methods). For each species, we sampled RNA from whole brains of adults for the two main social phenotypes – adult reproductives (defined as mated females with developed ovaries, henceforth referred to as 'queens' for simplicity; see Methods and Supplementary Data 1) and adult non-reproductives (defined as unmated females with no ovarian development, henceforth referred to as 'workers'; see Methods). These were sequenced as individual pools to generate a worker sample and a queen sample (both made up of between 3 and 11 biological replicates per caste pool). We then constructed de novo brain transcriptomes and estimated gene expression among queens and workers for each species. Using these data we could reconstruct a phylogenetic tree of the Hymenoptera using single-copy orthologous genes (Orthofinder[38]), resulting in expected patterns of phylogenetic relationships (Supplementary Fig. 1). This dataset allows us to test the extent to which the same molecular processes underpin the evolution of social phenotypes of different forms of sociality, as extant proxies in the major transition to superorganismality in wasps (Aim 1).

### Aim 1: Is there a shared genetic toolkit for caste among species across the major transition from non-superorganismality to superorganismality in Vespid wasps?

We found several lines of evidence for a shared genetic toolkit for caste across the wasp species using two different analytical approaches (Hypothesis 1).

Firstly, we found evidence that caste explains gene expression variation, after species-normalisation. The main factor explaining gene expression variation (using single copy orthogroups) was species identity, and to some extent subfamily, as all the Polistinae clustered tightly (Fig. 2a). However, since we are interested in determining whether there is a shared toolkit of caste-biased gene expression across species, we needed to control for the effect of species in our data. To do this, we performed a between-species normalisation on the transcript per million (TPM) score, scaling the variation of gene expression to a range of −1 to 1 (see Methods). After species-normalisation, the samples separate mostly into queen and worker

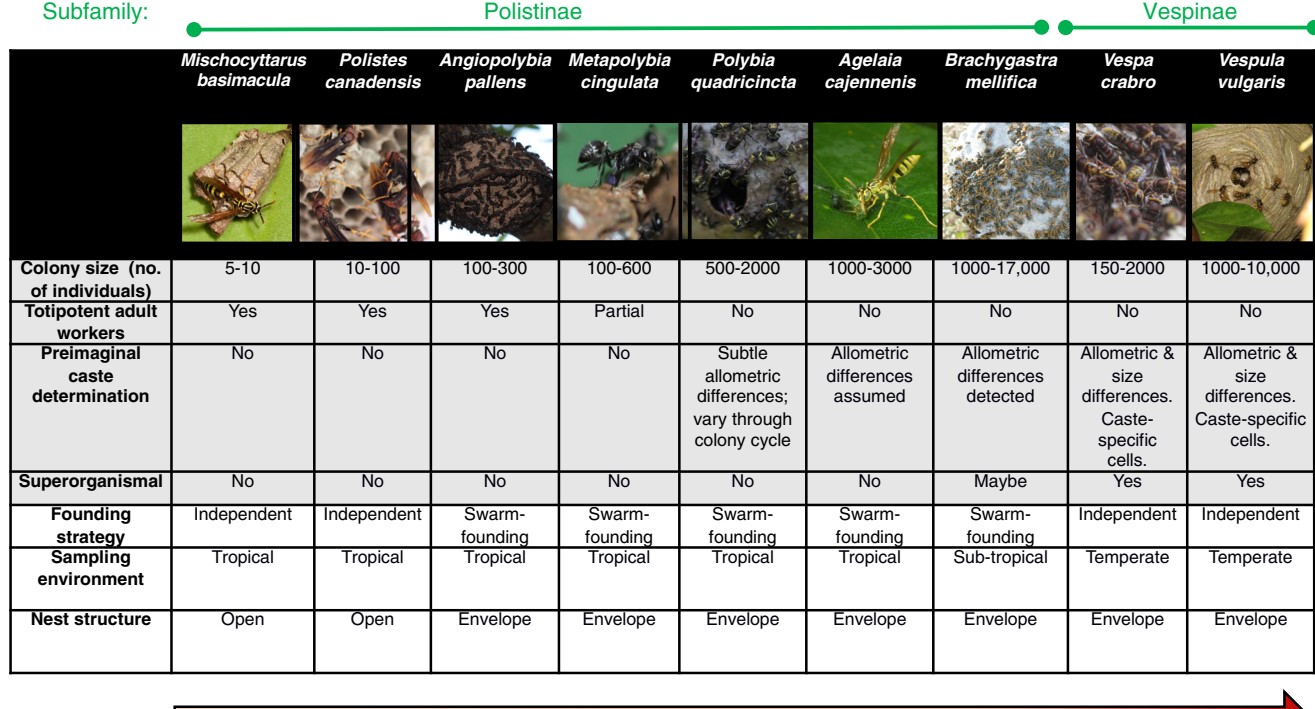

| Subfamily: | Polistinae | | | | | | | Vespinae | |
|---|---|---|---|---|---|---|---|---|---|
| | *Mischocyttarus basimacula* | *Polistes canadensis* | *Angiopolybia pallens* | *Metapolybia cingulata* | *Polybia quadricincta* | *Agelaia cajennenis* | *Brachygastra mellifica* | *Vespa crabro* | *Vespula vulgaris* |
| Colony size (no. of individuals) | 5-10 | 10-100 | 100-300 | 100-600 | 500-2000 | 1000-3000 | 1000-17,000 | 150-2000 | 1000-10,000 |
| Totipotent adult workers | Yes | Yes | Yes | Partial | No | No | No | No | No |
| Preimaginal caste determination | No | No | No | No | Subtle allometric differences; vary through colony cycle | Allometric differences assumed | Allometric differences detected | Allometric & size differences. Caste-specific cells. | Allometric & size differences. Caste-specific cells. |
| Superorganismal | No | No | No | No | No | No | Maybe | Yes | Yes |
| Founding strategy | Independent | Independent | Swarm-founding | Swarm-founding | Swarm-founding | Swarm-founding | Swarm-founding | Independent | Independent |
| Sampling environment | Tropical | Tropical | Tropical | Tropical | Tropical | Tropical | Sub-tropical | Temperate | Temperate |
| Nest structure | Open | Open | Envelope | Envelope | Envelope | Envelope | Envelope | Envelope | Envelope |

**Fig. 1 | Social wasps as a model group.** The nine species of social wasps used in this study, and their characteristics in social complexity and life history. The Polistinae and Vespinae are two subfamilies comprising 1100+ and 67 species of social wasp respectively, all of which share the same common nonsocial ancestor, an eumenid-like solitary wasp[92]. The Polistinae are an especially useful subfamily for studying the major transition as they include species exhibiting simple group living (with <10 individuals) of totipotent relatives, as well as species with varying degrees of more complex forms of sociality, with different colony sizes, levels of caste commitment and reproductive totipotency[93]. The Vespinae include the yellow-jackets and hornets, and are all superorganismal, meaning caste is determined during development in caste-specific brood cells; they show species-level variation in complexity, in terms of colony size and other superorganismal traits (e.g. multiple mating, worker policing)[94]. Ranked in order of increasing levels of social complexity, from simple to more complex, these species are: *Mischocyttarus basimacula*, *Polistes canadensis*, *Angiopolybia pallens*, *Metapolybia cingulata*, *Polybia quadricincta*, *Agelaia cajennensis*, *Brachygastra mellifica*, *Vespa crabro* and *Vespula vulgaris* (see Methods for further details of species choice). Where data on evidence of morphological castes was not available from the literature, we conducted morphometric analyses of representative queens and workers from several colonies per species (see Methods; Supplementary Data 8). With respect to life-history traits, all social wasps are generalist predators and so they share similar diets[56]. Other traits vary across species: e.g. some species founding nests as a single queen ('independent founding') whilst others found new nests as a group of queen(s) with workers ('swarm founding'); some species have open nests whilst others have an envelope. All Polistines included in this study are tropical, and all Vespines are temperate. The latter two traits are confounded by level of sociality and thus their influence on gene expression could not be disentangled. Image credits: *M. basimacula* (Stephen Cresswell). *A. cajennesis* (Gionorossi; Creative Commons); *V. vulgaris* (Donald Hobern; Creative Commons). *V. crabro* (Patrick Kennedy); *P. canadensis*; *M. cingulata*, *A. pallens*, *P. quadricinta*, (Seirian Sumner), *B. mellifica* (Amante Darmanin; Creative Commons).

phenotypes in the top two principal components (PC; Fig. 2b). This suggests that subsets of genes (a potential toolkit) are shared across these species and are representative of caste differences. However, there were outliers: *Brachygastra* did not cluster with any of the other samples (despite correct PC 1 queen direction); *Agelaia* showed little caste-specific separation and in the opposite direction to the other species (Fig. 2b). These two outlier species do not share the same caste-specific patterns as the other species. Neither sequence abnormalities nor contamination can explain why these species are outliers (see Supplementary Data 1).

Next, we used inferred lists of orthologs allowing some flexibility in numbers of isoforms (maximum of three isoforms, taking the most highly expressed) for each orthogroup and missing data (maximum of two species), allowing larger matrices of genes to be tested for differences between castes (Supplementary Data 2). Using this approach we detected 57 orthologous genes with four or more caste-biased species [queen or worker biased] (Fig. 3a; Supplementary Data 3) and 353 orthologous genes (Supplementary Data 3) with caste-biased expression in at least two of nine species in the same direction (either queen- or worker-biased only). There was overrepresentation of pheromone, chitin/odorant binding and muscle-related gene ontology

(GO) terms using these two cut-offs (Fig. 3b, c; Supplementary Data 3). No ortholog showed consistent caste-biased differential expression across all nine wasp species (Fig. 3a; Supplementary Data 3; using unadjusted <0.05 p values). Despite this, notable signatures of caste regulation were apparent across the species, with the exception of *Brachygastra*. For example, orthogroup OG0001418 was upregulated in queens from eight of the nine species and is predicted to belong to the vitellogenin gene family (using the *Metapolybia* protein sequence to represent the orthogroup), a transcriptional target and mediator of the insect physiology-regulator juvenile hormone (JH). JH is known as a pleiotropic master regulator of social traits, such as aggression, foraging behaviour and reproduction; in both Vespinae and Polistinae JH has been shown experimentally to regulate reproduction and is thought to be an honest fertility signal[39,40]. A second gene of interest that was consistently queen-biased was oocyte zinc finger 22. Zinc finger proteins have diverse functions but broadly they regulate gene expression by binding to DNA or RNA and controlling transcription; they are caste-biased in a range of social insects and have also been identified as targets for positive selection in social evolution[22] and so could be a key transcription factor driving wide expression patterns among castes across our species. Finally, we also could show that most

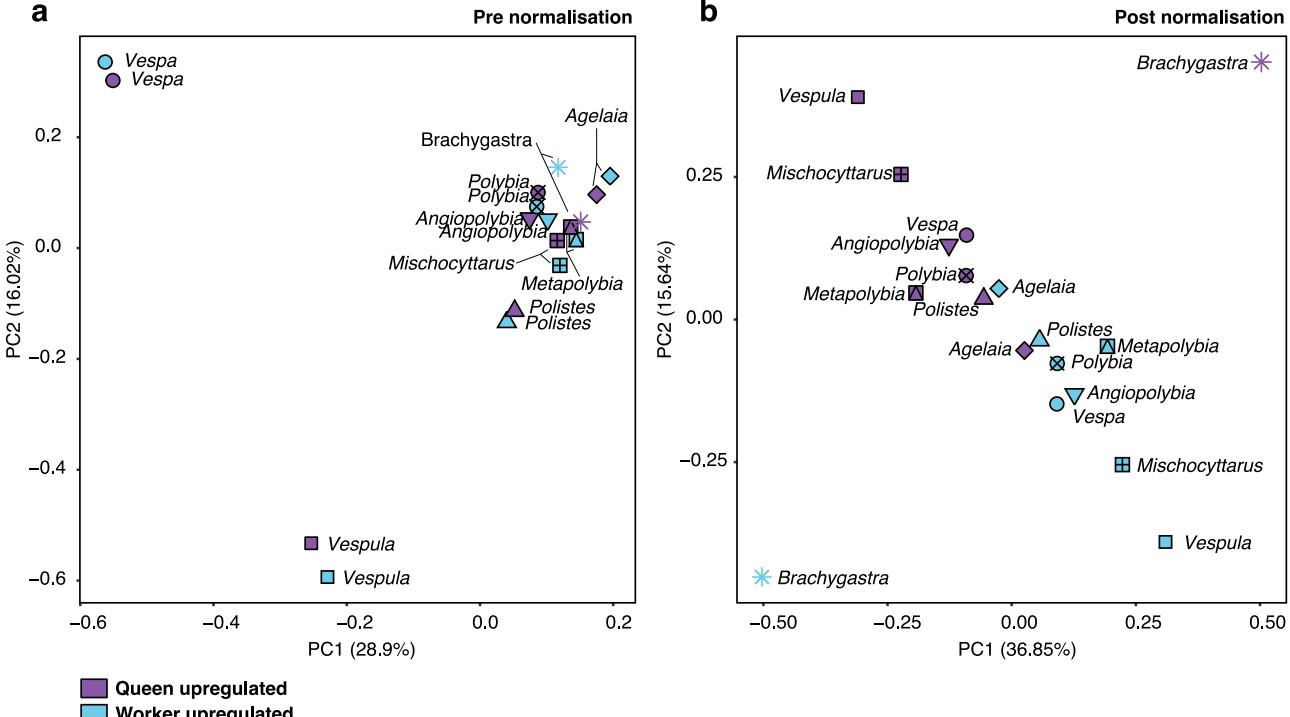

**Fig. 2 | Principal component analyses of orthologous gene expression before and after between-species normalisation. a** Principal component analyses performed using log2 transcript per million (TPM) gene expression values. This analysis used single-copy orthologs (using Orthofinder), allowing up to three gene isoforms in a single species to be present, whereby we took the most highly expressed to represent the orthogroup, as well as filtering of orthogroups which have expression below 10 counts per million. **b** Principal component analysis of the species-normalised and scaled TPM gene expression values using same filters as (**a**). Caste denoted by purple (queen) or blue (worker). Each species has a different shape/symbol. The percent of variation explained by the first two principal components is listed in brackets.

pair-wise differential genes overlap significantly, except some comparisons to *Agelaia* (Supplementary Note 1, Supplementary Fig. 6).

Secondly, we found evidence that a toolkit of many genes with small effects predicts caste across different forms of sociality in wasps. Conventional differential expression analyses (e.g. edgeR) require a balance of *P* value cut-offs and fold change requirements to reduce false-positive and false-negative errors[41]. Therefore, consistent patterns of many genes with smaller effect sizes may be missed when applying strict statistical measures. Support vector machine (SVM) approaches use a supervised learning model capable of classifying samples based on the effects of multiple independent variables (in our case: genes) to known response variables (in our case: castes, queens/workers)[42]. SVM approaches can detect subtle but pervasive signals in differential expression between states/conditions (e.g. for classification of single cells[43,44], cancer cells[45] and in social insect castes[46]), and reduce this information down to a single dimension (from 0 to 1). We used this approach to train SVM models, coding caste for each species as either 1 for queen and 0 for worker, and then testing whether we can successfully classify caste identity for a species not used to train the model; accurate classification of samples as queens (1) or workers (0) based on their global transcription patterns would be evidence for a genetic toolkit underpinning social phenotypes.

To run the SVM model we additionally used 'feature selection', which is a commonly used method of reducing the number of genes used for the predictions. In our model, we used linear regression to sort the genes into those relating to caste differences (in Figs. 4 and 5, we plot this on the x axis, where the SVM runs using 99 to 1% of the data remaining, which shows the percentage of total genes used in each svm model after feature selection).

After testing the SVM classification error and parameter settings (Supplementary Data 2, Supplementary Fig. 2), we conducted a "leave-

one-out" SVM approach to predict the caste in the species (queen) left out. The classifier estimation score ranges from 0 to 1, with an estimation closer to '0' meaning the signal is more worker-like and '1' for more queen-like (a score close to 0.5 means the SVM was unable to classify the sample). We applied three different iterations of the leave-one-out approach using different Orthofinder conversion tables (Supplementary Data 2). Our first approach used the "true-single-copy" orthologues that could be detected with sufficient expression (Fig. 4a-left; Supplementary Data 4) across all nine species (*n* = 1221 genes). After feature selection (based on linear regression, with more features removed from left to right; see Methods) the top 259 genes (<0.05 from linear regression representing around 20% of the genes after feature selection) could predict the queen caste correctly in seven of the nine species (Fig. 4a- left; with > = 0.6 classification estimate in the queen sample within the top 20% of genes); the two outlier species (*Agelaia* and *Brachygastra*) showed generally lower queen estimations (<0.5) and had previously been highlighted as outliers (Fig. 2 & 3). We explored two other orthology lists in order to explore whether, if the single-copy ortholog rule was relaxed and more orthogroups were included, caste estimations would remain accurate: in one model we allowed up to 3 isoforms per gene (Fig. 4a- middle: Supplementary Data 4; *n* = 1831 genes), and in the third we allowed up to two out of nine species to have a missing gene representative of the orthogroup, plus up to three isoforms per gene (Fig. 4a- right: Supplementary Data 4; *n* = 5536 genes). Even with these relaxed conditions, caste was classified accurately for all species except for the same aforementioned outliers. This suggests that large numbers of genes with caste-biased expression exist across social wasps.

We found some level of functional enrichment among the predictive genes that remained in the models after feature selection. There were consistent patterns of enrichment for GO terms in the

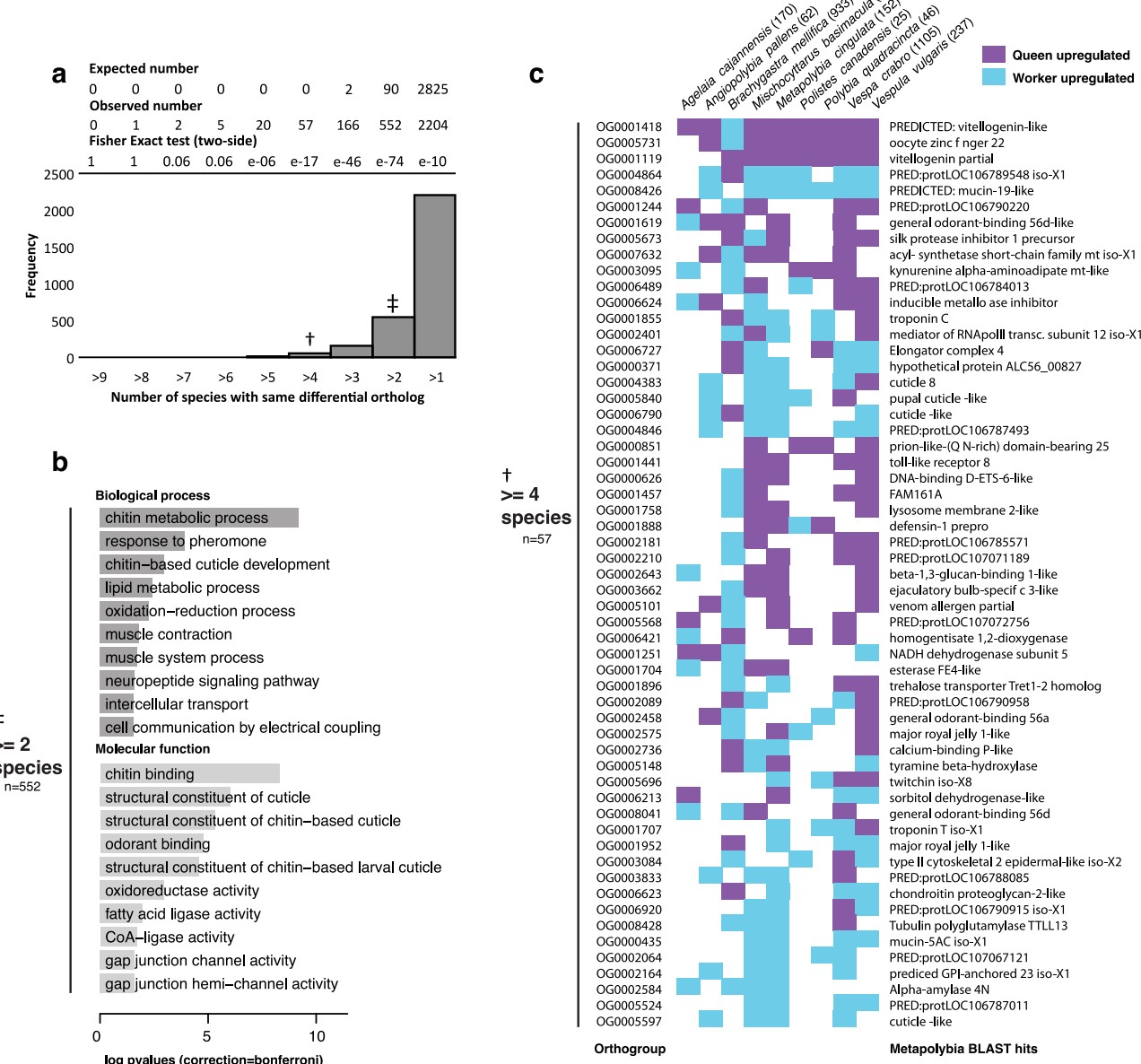

**Fig. 3 | Overlap of differential caste-biased genes (queen vs worker) and their functions. a** Frequency of caste DEGs in multiple species (observed, bar-plot). Showing numbers of orthologous genes that are differentially expressed in 1 up to 9 species. At the top we show the average "expected" numbers of DEGs to overlap, based on 1000 permutations with random genes, "observed" and Fisher two-sided P values for each expected/observed frequency indicate that the observed overlap in DEGs are more than expected by chance. Symbols represent datasets used in the gene ontology analysis (**b**) and heatmap (**c**, to right). For the full list of genes, see Supplementary Data 3. **b** A histogram of overrepresented gene ontology terms of genes found differentially expressed in at least two out of the nine species (n = 562

genes; either queen or worker upregulated; without needing the DEG to be in same caste upregulated direction), using a background of all genes tested across the nine species. P values are single-tailed and Bonferroni corrected. **c** Heatmap showing the differential genes that are caste-biased in at least four species (identified using edgeR) using the orthologous genes present in the nine species. In cases where three isoforms exist for a single species orthogroup, only the isoform with greatest expression was considered, plus at least 2 species could have missing gene data for each orthogroup. Listed for each species (in brackets), is the total number of orthologous differentially expressed genes per species. *Metapolybia* Blast hits are listed along (right) with orthogroup name (left).

two more relaxed models (Fig. 4b- middle & 4b- right), with genes enriched for mostly ionic transport and visual/sensory perception (Supplementary Data 4). Expression of the top 50 genes (from linear regression; chosen from Fig. 4a- left) allow us to identify some of the most important genes (Fig. 4c), some of which were previously been identified as associated with caste differentiation in social insects (e.g. vitellogenin; zinc finger (as mentioned earlier) and ubiquitin). Others have not been associated with caste previously but may have relevant functions in regulating caste; for example, striatin-4 (OG0005432) is involved in calmodulin binding and thus may play a

role in activating (or inactivating) calcium-binding; adenosylhomocysteinase (OG0002120) is one of the most conserved proteins in living organisms and facilitates local transmethylation, thus it may be important in active transcription and epigenetic processes[47]. Interestingly, the vast majority of the putatively important genes are downregulated in queens, and upregulated in workers, suggesting that perhaps the transcriptomic demands of worker behaviour may be more diverse than those involved in being a queen, whilst queens express a more restricted molecular toolkit as expected if they are specialised in reproduction.

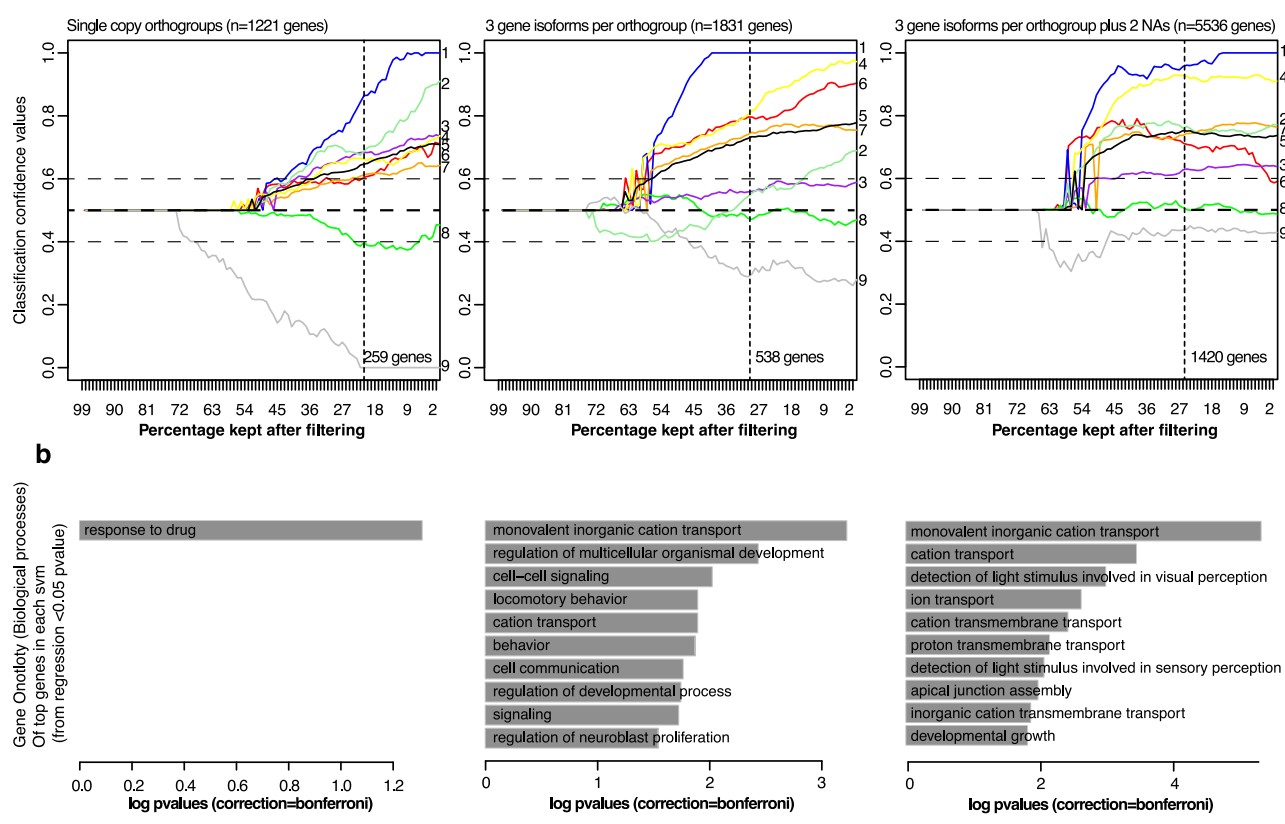

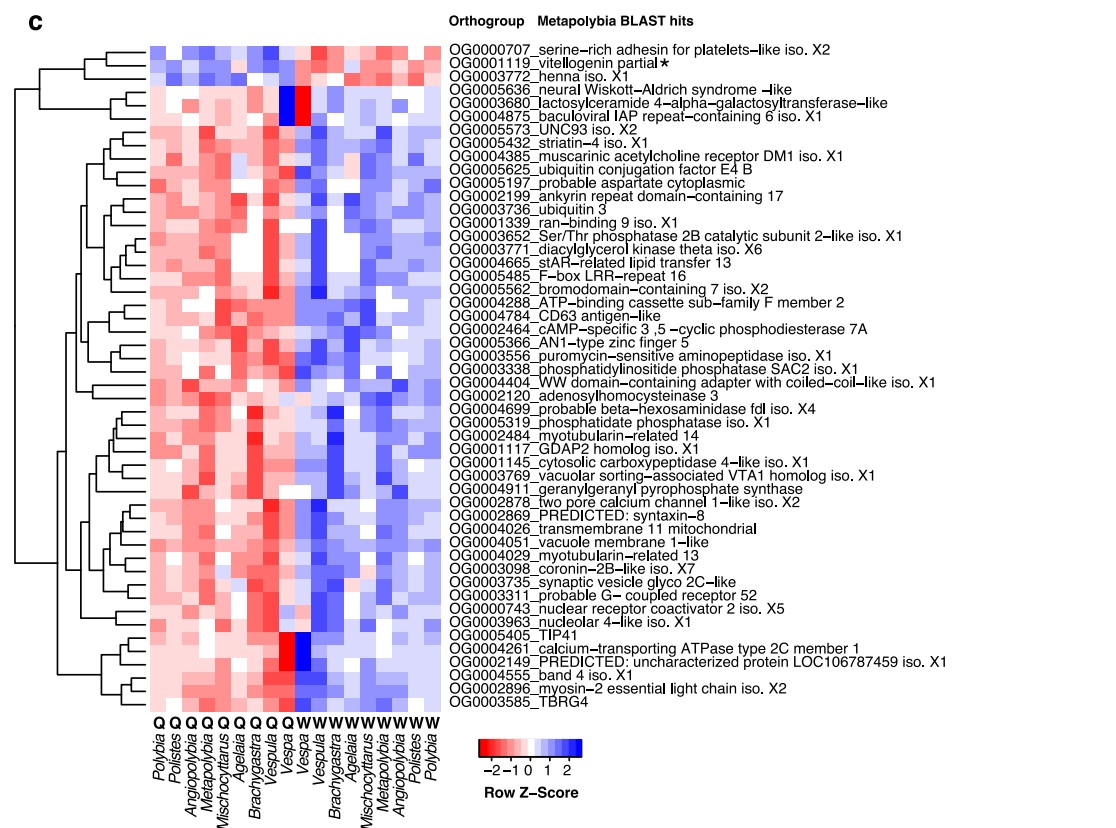

**Fig. 4 | A genetic toolkit for social behaviours across eusocial wasps. a** Change in certainty of correct caste classifications through progressive feature selection using Support Vector Machine (SVM) approaches. Models were trained on eight species and tested on the ninth species, where we instruct the model as to which sample represents the queen and which the worker for the eight species. Features (a.k.a. orthologous genes) were sorted by linear regression (x axis) with regard to caste identity within the 8 training species, beginning at 99% where almost all genes were used for the predictions of caste, to 1% where only the top one percent of genes from the linear regression (sorted by *P* value) were used to train the model. '1.0' equates to high queen classification certainty. This test was repeated using various filters, showing: LEFT: use of only 1 to 1 orthogroups, where no gene isoforms or missing values were accepted into the model, MIDDLE: allowing three gene isoforms (using the most highly expressed as the representative) and RIGHT: allowing three gene isoforms and missing values (NAs; up to 2 species with no ortholog), filling in with non-caste biased gene values for queen and worker. Horizontal guide markers are placed at 0.4, 0.5 and 0.6 respectively indicating the confidence score for classifying queens, and the vertical dotted line marks the percentage at which the regression score becomes <0.05. **b** Histograms indicate the Bonferroni-corrected enriched gene ontology (GO) terms for the top orthologous genes from the corresponding model above (linear regression *P* value < 0.05; shown as dotted line in upper plots), tested against a background of all genes used in the corresponding SVM model (i.e. with a single gene representative for all species in the test). *P* values are single-tailed. **c)** Heatmap of the top 50 genes after sorting by linear regression, in the SVM with 3 merged isoforms and 2 NAs; showing species-normalised gene expression levels in the nine species for queen (Q) and worker (W) samples (where red indicated down-regulation and blue up; row Z-score). Orthogroup name and top *Metapolybia* blast hit are listed to the right.

We compared the genes and GO terms obtained from the SVM and edgeR methods (Fig. 3c). There was little overlap in enrichment of GO terms between the two datasets. Moreover, there was less overlap than expected by chance between the 1420 SVM predictor genes and the 353 genes identified as differentially expressed (edgeR; <0.05 P value): in comparisons that included at least two species in the same direction, only 35 genes overlapped (Hypergeometric: 1.89e-14, under-enriched 2.59 fold; Supplementary Data 4). Of the genes that are present in both sets, some have previously been identified as having relevance to social evolution and caste differentiation; these include Vitellogenin (as mentioned earlier;[48] two protein isoforms: OG0001119 and OG0001418), a zinc finger gene (as mentioned earlier; OG0005731) and a gonadotrophin-releasing hormone, which is related to the regulation of reproduction. ATP-synthase has caste-specific expression in the weaver ant[49] (OG0006255 and OG0006612) and esterase E4-like (OG0000783) is upregulated in young honeybee queens compared to nurses at the proteomic level[50]. There are also other genes of interest, which to our knowledge have not previously associated with caste, including Toll-like receptor 8 (OG0001441) (see Supplementary Data 4). Finally, 40% of the overlapping genes (*n* = 14) are uncharacterised proteins, suggesting a role for potentially novel genes; novel (or taxon-restricted) genes have been previously highlighted for their potential role in social evolution across bees and wasps[15].

## Aim 2: How does level of social complexity, phylogeny and life-history influence the molecular basis of castes?

To explore whether there are any finer-scale differences among the different levels of social complexity (Hypothesis 2)[24], we trained an SVM model using *Mischocyttarus, Polistes, Angiopolybia* and *Metapolybia* as representatives of the simpler societies in the major transition (see Fig. 1), and tested how well this gene set classified castes for the four species representing the more complex societies in the major transition (*Polybia, Agelaia, Vespa* and *Vespula*; see Fig. 1). *Brachygastra* was excluded due to its poor performance overall (see Fig. 4) and to balance sample sizes in training and test sets. If castes in the test species classify well, this would suggest that the processes regulating castes in the simpler societies are equally important in the more complex societies (i.e. there is no specific caste toolkit for social behaviour in simple societies, which is absent (putatively lost) in more complex societies). Conversely, if the test species do not classify well, this would suggest that there are distinct processes regulating caste in the simpler societies that are less important in more complex forms of sociality.

When training the model on the four species with simple societies, we found that both vespines (*Vespa* and *Vespula*) had good classification estimates at around 0.8 (Fig. 5a), comprising around 800 genes after feature selection (*p* values < 0.05 [vertical dotted line in Fig. 5a]). Similarly, *Polybia* also classified well, though with lower confidence (~0.78). *Agelaia* could not classify castes correctly (~0.55); however,

this species was identified as an outlier in our first analyses and this may reflect some facet of their biology (e.g. their intermediate form of social complexity – see Fig. 1), or an anomaly with the data that we were unable to detect (see Supplementary Data 1). Considering these four tests together, a set of 277 gene orthologs were found (Supplementary Data 5; *p* value < 0.05 in all four species tests), that represent a core set of genes that classify caste correctly in complex societies. Based on these species, these results suggest that the genetic toolkit for simple societies is well conserved in the more complex societies that we sampled, although future work should confirm this by expanding the repertoire of species.

We next performed the reciprocal test by training the SVM using the four species with the more complex societies (*Polybia, Agelaia, Vespa* and *Vespula*) and tested it on the four species with simpler societies (*Mischocyttarus, Polistes, Metapolybia* and *Angiopolybia)*. The predictive gene set for the more complex societies was able to classify castes for the simpler societies albeit with lower overall levels of success (Fig. 5b - *Polistes* -0.65, *Angiopolybia* -0.7, *Metapolybia* -0.7, *Mischocyttarus* -0.8), and with a significantly smaller gene set. After feature selection, only around 350 genes were found in each of the four tests (less than half that found in the first test: ~800), and 289 were found in all four tests, describing castes in these more complex societies (Supplementary Data 5). The lower overall success of the gene set trained on the more complex species in classifying castes in species from the simpler societies suggests there may be different (additional) processes regulating castes at the more complex levels of sociality.

To explore this idea further, we tested whether there was significant overlap among the three putative toolkits, by comparing the identities of genes predictive of 'simpler societies' (Fig. 5a; *n* = 277), 'more complex societies' (Fig. 5b; *n* = 289), and 'full spectrum' (Fig. 4a-Left; *n* = 259) toolkits. We found significant overlap between the combined genes in the simpler and the more complex 'toolkits' with the full spectrum set (Fig. 5c grey vs pink; blue vs pink), but not between the simpler and more complex societies (grey vs blue; see Fig. 5c for statistics). This suggests that there may be distinct differences in the genes regulating castes at different levels of social complexity. We found further suggestive evidence for this at the putative functional level: there were no consistent patterns of GO terms across the two toolkits, with only the complex society toolkit having significant levels of enrichment after correction (Fig. 5d- mostly cation/ion transport; Supplementary Data 5).

Next, we explored whether the genes regulating castes varied among subfamilies (Hypothesis 3). Using the same reciprocal SVM approach, we found little evidence that subfamily influences the molecular processes regulating caste. Using the seven polistines as the training set, queens and workers in the two vespines mostly classified with a confidence of 1 with less than ~35% of genes removed after feature selection, similar to our previous classifications (Supp. Figure 3a, Supplementary Data 6 [tab

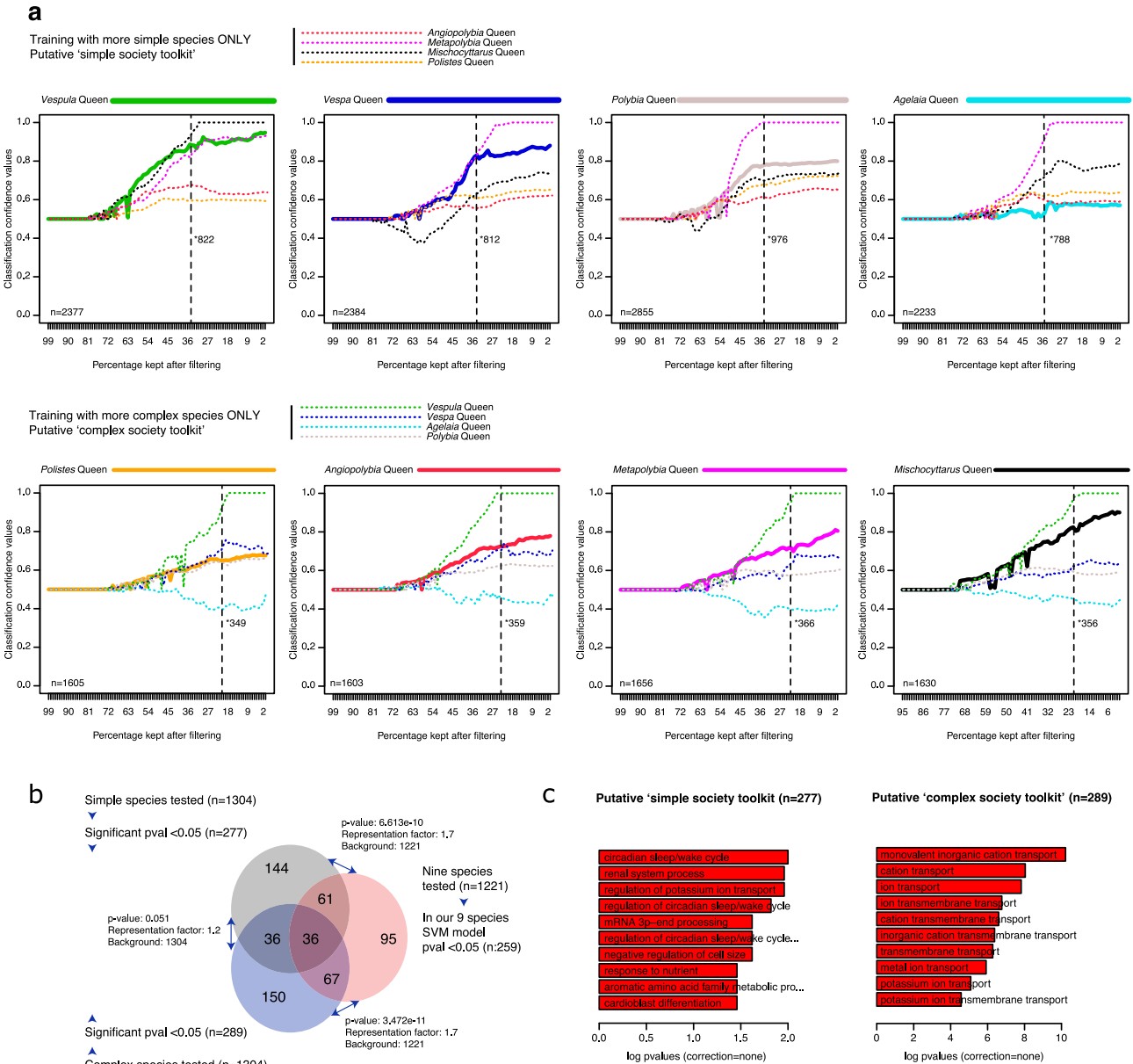

**Fig. 5 | Testing for the presence of a distinct 'simple society toolkit' and a 'complex society toolkit'. a** Using either the four species with the simpler or more complex societies, we trained SVM models with information as to which sample represents the queen and worker for the four species. We used a progressive filtering of genes (based on linear regression; the same method used in Fig. 4), to assess how well we could predict caste using more informative genes. UPPER panel: training with the four simpler species, and tested on each of the four more complex species. LOWER panel: training with the four more complex species, and tested on each of the four simpler species. Classification estimates of being a queen from 0 to 1 is plotted for each species across the progressively filtered sets. The total number of orthologous genes used in each SVM model are shown (bottom left of each panel); the total number of these genes that were significant predictors of caste (*p* value < 0.05) is listed at the dotted line. For each test (pair of Queen/Worker in each

species), the SVM model was run using genes with only 1 homologous gene copy per species. **b** Overlap of significant genes among the three putative toolkits, comparing those genes for 'simpler species' (Fig. 5a UPPER panel), 'more complex species' (Fig. 5a LOWER panel), and 'full spectrum' (Fig. 4a- Left) toolkits. For each experiment, the number of orthogroups tested and the total number that were significant after linear regression are shown. Inside the Venn diagram are the numbers of these significant genes shared (or not) between the groups. Significant overlap is shown using hypergeometric tests (one-tailed). The colours represent the genes that predict caste in: the more complex societies (blue), the simpler societies (grey) and across the full spectrum (pink). **c** Enriched gene ontology terms (TopGO) using a background of all genes tested (*n* = 1304) for the two putative toolkits (focal sets *n* = 277 and *n* = 289), using an uncorrected single-tailed *p* values.

'Vespula_Polistinae' and 'Vespa_Polistinae']). The reverse of this test could not be performed due to low sample sizes for a vespine training set. Despite this, we conducted further tests using five Polistines species (of the genera *Metapolybia*, *Mischocyttarus*, *Polistes*, *Polybia* and *Angiopolybia*; removing the weakly supporting species *Agelaia*) to see if training solely within their own subfamily would improve or worsen the classification scores.

Predictions were generally similar or worse than when all remained species are used from training (except *Metapolybia*). This suggests that the signal of caste is not particularly strong in the Polistines and that this group does not have a specific toolkit isolated from the Vespines (that would be expected to improve classifications); however, more data would be needed to fully test this hypothesis. Thus, with the data currently available, these results suggest that

the genes regulating castes are shared across these two subfamilies, with little evidence for subfamily-specific toolkit genes.

Vespid wasps share many life-history traits (e.g. diet), but are contrasting in others. One contrasting trait, which was not confounded by level of social complexity (see Fig. 1) is their mode of colony founding. Some species form new colonies alone ('independent founders') whilst others form new colonies as a swarm containing queens and workers ('swarm founders'). Founding behaviour is independent of level of complexity: e.g. the species with the simplest (e.g. *Polistes* and *Mischocyttarus*) and the most complex (*Vespa* and *Vespula*) societies are all independent founders; equally the swarm-founders (*Angiopolybia, Metapolybia, Agelaia*, and *Polybia*) exhibit a spectrum of social complexity (see Fig. 1). It is possible, therefore, that the molecular basis of caste reflects this key life-history trait, and not just social complexity (Hypothesis 4). We found some evidence to support this. The gene set obtained from an SVM trained to classify castes in the swarm-founders was very successful in classifying castes for the independent founders, with classifications between 0.7–1 for all test species. Conversely, the reciprocal analysis performed poorly: the gene set obtained from an SVM trained on the independent-founders was unable to predict castes at >0.7 for three of the four species (Supplementary Fig. 4; Supplementary Data 6 [Swarm/Independent]); however, without greater replication, the contribution of life history traits to the signal of caste across the Hymenoptera remains unclear. In addition, we also found some evidence of elevated rates of protein evolution among swarm-founders specifically (see Supplementary Note 2), suggesting that life history could involve additional molecular innovations.

Further support for there being different molecular processes shaping caste transcription at different levels of social complexity was found in the patterns of gene evolution (Hypothesis 5). Previous studies found that rates of protein evolution are high in more complex societies and specifically among the caste-biased genes[18,19]. We tested both these predictions.

We found no support for the prediction that higher rates of evolution would be found in the more complex societies (Supplementary Fig. 5a, b). To the contrary, overall the highest rates of $dN/dS$ were in the four species with simpler societies, with an average of 0.1639218 across all orthogroups, and significantly higher than the five species with more complex societies (average 0.1015024, Wilcoxon test W = 307318, $p$ value < 2.2e-16; Supplementary Fig. 5c). We dissected this further within the Polistinae: species with the simpler societies (*Mischocyttarus, Polistes*) had higher and significantly different rates of evolution than those with more complex societies (*Polybia, Brachygastra, Agelaia*) (*dNdS* mean of orthogroups in foreground branch with simple species = 0.180918; with complex species = 0.1327845; Wilcoxon test W = 119615, $p$ value = 2.824e-11). Six orthogroups showed positive selection in only the simpler Polistines (chi-square test p < 0.05, dN/dS >1): these included genes related to female-specific gene expression in ants[51], ant venom[52] and social behaviour in zebrafish[53] (Supplementary Data 7, tab ST15). Four orthogroups showed positive selection in only the complex Polistines, none of which had been linked previously to social behaviour (Supplementary Data 7, tab ST16). Across both lineages (polistines and vespines), one orthogroup was under positive selection among the complex species (Supplementary Data 7, tab ST14), suggesting lineage effects and the need for wider taxonomic sampling of vespines.

We found little support for the prediction that caste-biased genes (specifically worker-expressed) would be rapidly evolving. Of the 11 significant orthogroups identified above, only one was differentially expressed with respect to caste in two species (OG0004867, uncharacterized protein LOC106789623), and four other orthogroups (OG0003362: UMP-CMP kinase isoform X1; OG0003491: uncharacterized protein LOC107067502; OG0003159: lysozyme c-1-like; OG0003184: doublesex isoform X1) were differentially expression in just one species (Supplementary Data 7, tab ST27).

Taken together, these results provide evidence that the molecular processes regulating caste differentiation vary depending on the level of social complexity. The finding of more rapid rates of protein evolution in the simpler societies (Supplementary Fig. 5c) was unexpected since studies of gene evolution in social bees have found the higher rates of evolution in the more complex societies[6–8,13,17,19,20].

## Discussion

Major transitions in evolution provide a conceptual framework for understanding the emergence of biological complexity. Discerning the processes by which such transitions arise provides us with critical insights into the origins and elaboration of such complexity. In this study, we explored the evidence for key hypotheses on the molecular bases of social evolution by analysing caste transcription in nine species of wasps. As predicted, we find evidence of a shared genetic toolkit across the spectrum of social complexity in wasps;[25] importantly, using machine learning we reveal that this toolkit likely consists of many hundreds of genes of small effect (Fig. 4). However, in sub-setting the data by level of social complexity, lineage and a key life-history trait (colony founding), several insights are revealed. Firstly, there appears to be a putative toolkit for castes in the simpler societies that largely persists across the major transition, through to superorganismality. Secondly, different (additional) processes appear to become important at more complex levels of sociality[24]. Finally, we discuss apparent differences among lineages, highlighting the importance of wider taxonomic consideration and replication in the quest to understand the molecular basis of sociality and the nature of the major transition. Taken together, we suggest that the concept of a shared molecular toolkit for sociality may be too simplistic and that future studies should consider the evolutionary impact of other traits on molecular processes, specifically life-history and ecology, alongside castes.

The discovery of a substantial set of genes that consistently classify caste across most of the species, irrespective of the level of social complexity was a key finding. The taxonomic range of samples used meant we were able to confirm that specific genes are consistently differentially expressed, with respect to caste, across the species. These patterns would be difficult to detect if only looking at a few species within a lineage or among species representing only a limited selection of social complexities. In addition to typical caste-biased molecular processes, we also identified that genes related to synaptic vesicles are different between castes; this is interesting as the regulation of synaptic vesicles affects learning and memory in insects[54]. To our knowledge, this is the first evidence of what may be a conserved genetic toolkit for sociality, from the simplest stages of social living through to true superorganismality.

The underlying assumption, based on the conserved toolkit hypothesis, has been that whatever processes regulate castes in complex societies must also regulate castes in simpler societies[7,25,55]. Alternative hypotheses posit that different molecular processes underpinning castes may differ at different levels of social complexity[24]. We found evidence supporting this latter idea. There may be fundamental differences discriminating superorganismal societies from non-superorganismal societies[24]. Intriguingly, the two species which classified least well (*Agelaia* and *Brachygastra*) exhibit intermediate states of social complexity between superorganismal and non-superorganismal, sharing traits of both (see Fig. 1). Further analyses from these kinds of societies are required to explore whether these patterns reflect a form of sociality complexity where the conserved molecular toolkit is not important (or is different); or perhaps it may reflect a pivotal stage in the major transition, involving a possible upheaval of the molecular processes regulating castes, where pre-imaginal caste commitment is partially apparent. These findings highlight the importance of examining different stages in the

major transition when attempting to elucidate its patterns and processes[8,10,21,35].

Dissecting the molecular toolkits by lineage (Polistinae and Vespinae) and colony founding categories (swarm-founding and independent founding), showed that these factors also influence the molecular toolkit. The Vespines performed better when trained with the Polistines than the other way round, and there were some small differences between the swarm and independent founders, highlighting the importance of taking lineage and life-history traits into consideration.

Phenotypic evolution is often accompanied by increased rates of gene evolution. Indeed, high rates of protein evolution have been detected in worker-biased genes of more complex societies like the honeybee, and some ants[13,18–20]. This makes sense as the worker caste (with altruism and partial/complete sterility) is the dominant phenotypic innovation in social evolution and has led to the suggestion that rapid gene evolution is expected in the most complex forms of sociality, where phenotypic specialisations are most pronounced[18,21]. We were surprised to find, therefore, higher rates of gene evolution in wasp species with simpler societies compared to more complex societies; these results are in stark contrast with the findings of a recent brain/head transcriptome study on social phenotypes from a range of social complexity in bees[8] where (in keeping with previous studies) rapid gene evolution was most apparent in complex societies. There are several interpretations of these contrasting results. First, the level of social complexity displayed by the simplest species in Shell et al. 2021[8] represents a much more rudimentary form of sociality compared to the simplest societies in our study, with many of their 12 species being only facultatively sociality (e.g. females of *Ceratina australensis* can nest either alone or in a small group) and some barely exhibiting any reproductive division of labour (e.g. *Ceratina calcarata*). Facultative sociality is rare in the vespid wasps, being largely limited to a few species of Stenogastrinae; the simplest Polistinae are represented in our sample and all are obligately social (meaning they always nest in a group). It is possible therefore, that the rapid gene evolution occurs only when a species is committed to group living, with a clear division of reproductive labour. An alternative explanation lies in ecology and life-history: bees and wasps differ fundamentally in how they provision brood, with wasps hunting prey[56] and bees collecting pollen. These findings add to the complexities of explaining the molecular basis of social evolution, and highlight a greater need to emphasis and account for variation in natural history, ecology and life-history and social behaviour[10,26,35,57].

Our study illustrates the power of SVMs in detecting large suites of genes with small effects, which largely differ from those identified from conventional differential expression analysis[46]. We advocate the use of the two methods in parallel: our conventional analyses suggested that metabolic genes appear to be responsible for the differences between castes, whereas the SVM genes were mostly enriched in neural vesicle transportation genes, which have not previously been connected with caste evolution. However, these models are only as good as the initial data used to train them. Our study suffers from a few limitations, which are important to expand upon. Firstly, the sample size (number of species) is good for a sociogenomics study, but relatively low for machine learning studies; SVMs are generally used on very large datasets such as clinical trials in the medical sciences[58]. Our models did perform well, as evidenced from our exploration of learning curves and predicted errors (see Fig. 4a, Methods & Supplementary Data 9) and our analysis examined gene expression data with thousands of features; however, future analyses would be more robust by using more species in the training datasets. Indeed, we observed reduced performance in our model predictions when fewer species were included in the training set. Secondly, by comparing across multiple species, we can only train our model on genes that have a single representative isoform per species in each separate test. This reduces the numbers of genes we can test in each SVM model, especially where more distantly related species are included. We overcame this limitation by merging gene isoforms within the same orthogroup (potential gene duplications); yet this comes with some additional costs as some genes are discarded in this process. Third, genomes are not available for most of the species we tested; our measurements are based on de novo sequenced transcriptomes, which potentially contain misassembled transcripts, which could reduce the ability to find single-copy orthologs across species. For these reasons, the numbers of genes detected in our putative toolkit for sociality is likely to be conservative and modest (potentially by several fold). Fourth, although we sampled as widely as we could from different colonies within each species, there may be inherent differences associated with the biology of a particular species that may not be entirely accounted for due to the difficulty of getting hold of little studied, hard-to-find insects as used in our study. A key example is stage in the colony cycle which was accounted for in some species by sampling across multiple colonies (e.g. *Metapolybia; Polistes*) but not in others (e.g. we were only able to obtain one colony of *Agelaia)*. Fifth, we lacked true biological replicates of the majority of species, which prevents us from concluding with confidence the patterns in individual species. This could also mean that an individual with highly biased gene expression could distort the expression profile of the group it represents (queen or worker), and that by pooling several indviduals into a single replicate, we lose that information and thus cannot test for individual-level bias. Finally, our study focused on brain tissue only; although gene expression in the brain is dynamic and responsive to the social environment[37], brain tissue tends to have a lower number of highly expressed genes relative to other tissues and slower rates of gene evolution[17]. Future studies might consider a wider range of tissue types (e.g[33].) with higher levels of replication within and among species. These limitations apply to many studies to date, due to the difficulty and expense of obtaining high-quality genomic data for specific phenotypes for non-model organisms.

In conclusion, our analyses of brain transcriptomes for castes of social wasps suggest that the molecular processes underpinning sociality are conserved throughout the major transition to superorganismality. However, different (putatively additional) processes may come into play in more complex societies. There may be fundamental differences in the molecular machinery that discriminates key points of innovation in the major transition; for example, the evolution of irreversible caste commitment (in superorganisms) may require a fundamental shift in the underlying regulatory molecular machinery[6,24]. Such shifts may be apparent in the evolution of sociality at other levels of biological organisation, such as the evolution of multicellularity, taking us a step closer to determining whether there is a unified process underpinning the major transitions in evolution.

## Methods

### Study species
Nine species of vespid wasps were chosen to represent different levels of social complexity across the major transition (Fig. 1). The simplest societies in our study are represented by *Mischocyttarus basimacula basimacula* (Cameron) and *Polistes canadensis* (Linnaeus); wasps in these two genera are all independent nest founders and lack morphological castes (defined as allometric differences in body shape, rather than overall size) or any documented form of life-time caste-role commitment[59–62]. They live in small family groups of reproductively totipotent females, one of whom usually dominates reproduction (the queen); if the queen dies she is succeeded by a previously-working individual[22]. As such, these societies represent some of the simplest forms of sociality, where caste roles are least well defined, and where individual-level plasticity is advantageous for maximising inclusive fitness. These simple forms of group living may represent the early stages in the major evolutionary transition to sociality[25,35].

The Neotropical swarm-founding wasps (Hymenoptera: Vespidae; Epiponini) include over 19 genera with at least 246 species[63], exhibiting a range of social complexity measures, from complete absence of morphological caste (preimaginal) determination to colony-stage specific morphological differentiation, through to permanent morphological queen-worker differentiation[64]. As examples of species for which there is little or no evidence of developmental (morphological) caste determination, we chose *Angiopolybia pallens* (Lepeletier) which is phylogenetically basal in the Epiponines[65,66] and *Metapolybia cingulata* (Fabricius)[65,66]. We confirmed the lack of clear caste allometric differences in *Metapolybia cingulata* as data were lacking (see Morphometrics methods (below) and Supplementary Data 8).

As examples of species showing subtle, colony-stage-specific caste allometry, we chose a species of *Polybia*. The social organisation of *Polybia* spp is highly variable, ranging from complete absence of morphological queen-worker differentiation[67]. *Polybia quadricincta* (Saussure) is a relatively rare (and little studied) epiponine wasp, which can be found across Bolivia, Brazil, Columbia, French Guiana, Guyana, Peru, Suriname and Trinidad[68]. Our morphometric analyses found some evidence of subtle allometric morphological differentiation in this species, but with variation through the colony cycle (Supplementary Data 8); this suggest it is a representative species for the evolution of the first signs of pre-imaginal caste differentiation.

Many species of the genera *Agelaia* and *Brachygastra* appear to show pre-imaginal caste determination with allometric morphological differences between adult queens and workers[65]. We chose one species from each of these genera as representatives of the most socially complex Polistine wasps. Although no morphological data were available for *Agelaia cajennensis* (Fabricius) all species of *Agelaia* studied show some level of preimaginal caste determination[65,69]. *Brachygastra* exhibit a diversity of caste differentiation;[65,70] our morphological analysis of caste differentiation *B. mellifica* (Say) confirms that this species is highly socially complex, with large colony sizes[65] and pre-imaginal caste determination resulting in allometric caste differences (Supplementary Data 8).

All species of Vespines are independent nest founders and superorganisms, with a single mated queen establishing a new colony alone and with morphological castes that are determined during development. However, some species exhibit derived superorganismal traits, such as multiple mating[71], which have likely evolved under different selection pressures to the major transition itself[72]. The European hornet, *Vespa crabro* (Linnaeus), exhibits the hallmarks of superorganismality (see Fig. 1) but little evidence of more derived superorganismal traits, such as high levels of multiple mating. Conversely, multiple mating is common in *Vespula* species, including *V. vulgaris* (Linnaeus) with larger colony sizes than *Vespa*[71], suggesting a more complex level of social organisation.

## Sample collection
All colonies used in this study were post-emergence, meaning they were well established, with brood of all life stages (eggs, larvae and pupae) and were actively producing adults. Where possible, we sampled from colonies representing different stages in the colony cycle, as caste differentiation can vary as the colony matures in some species (Supplementary Data 1). All species were collected in the same way, as whole colonies collected in situ in the wild at a time of day when foraging was actively taking place. *Metapolybia cingulata* (6 colonies), *Polistes canadensis* (3 colonies), *Agelaia cajennensis* (1 colony) and *Mischocyttarus basimacula basimacula* (3 colonies) were collected from wild populations in Panama in June 2013. *Brachygastra mellifica* (4 colonies) were collected from populations in Texas, USA in June 2013. *Angiopolybia pallens* (2 colonies) and *Polybia quadricincta* (2 colonies) were collected from Arima Valley, Trinidad in July 2015. *Vespa crabro* (4 colonies) and *Vespula vulgaris* (4 colonies) were collected from various locations in South-East England, UK in 2017. All colonies were collected in a similar way, with wasps picked from the nest and placed immediately onto dry ice or directly into RNA*later* (Ambion, Invitrogen) and stored at −20 °C until further use. Samples were ultimately pooled *within castes* for all analyses, such that each pool consisted of 3–6 individual brains from wasps sampled across 2–4 colonies to capture individual-level and colony-level variation in gene expression (see Supplementary Data 1). Samples of *M. cingulata, A. cajennensis, M. basimacula* and *B. mellifica* were sent to James Carpenter at the American Natural History Museum for species verification. *A. pallens* and *P. quadricincta* were identified by Christopher K. Starr, at University of West Indies, Trinidad and Tobago.

## Identification of queens and workers
Queens and workers were identified through examination of their reproductive tracts as this shown to be a reliable indicator of caste in social wasps. Abdomens were removed from all wasps in each colony and all females were dissected to determine reproductive status. Ovary development was scored according to[35,57] and the presence/absence of sperm in the spermathecae was identified to determine insemination. Inseminated females with developed ovaries were scored as 'queens'; non-inseminated females with no sign of any ovary development were scored as workers. Uninseminated wasps with developed ovaries are sometimes found in some of these species, especially the epiponines where they may be laying unfertilised (male) eggs (known as 'intermediates'); no wasps in this category were used in this study as we were specifically interested in comparing clear reproductive queens and non-reproductive workers across species. Callows (newly emerged wasps) were also excluded from analysis, recognised by their colour (dull cuticle, eye colour), lack of wing wear and/or their internal anatomy (callows typically have a chalky mass inside their abdomen).

## Morphometrics
Data on morphological differentiation among colony members (and thus information on whether pre-imaginal (developmental) caste determination was present) was lacking for *M. cingulata, P. quadricinta* and *B. mellifica*; therefore, we conducted morphometric analyses on these three species in order to ascertain the level of social complexity. Morphometric analyses were carried out using GXCAM-1.3 and GXCapture V8.0 (GT Vision) to provide images for assessing morphology. We measured 7 morphological characters using ImageJ v1.49 for queens and workers for each species. The body parts measured were: head length (HL), head width (HW), minimum interorbital distance (MID), mesoscutum length (MSL), mesoscutum width (MSW), mesosoma height (MSH) and alitrunk length (AL) (for measurement details, see[73]). Abdominal measurements were not recorded as ovary development could alter the size of abdominal measurements, therefore biasing the results. The morphological data were analysed to determine whether the phenotypic classification, as determined from reproductive status, could be explained by morphological differences. ANOVA was used to determine size differences between castes for each morphological characteristic. A linear discriminant analysis was also employed to see if combinations of characters were helpful in discriminating between castes. The significance of Wilks' lambda values were tested to determine which morphological characters were the most important for caste prediction. All statistical analyses were carried out using SPSS v23.0 or Exlstat 2018. Data and analyses given in Supplementary Data 8.

## RNA extractions
Individual heads were stored in RNAlater for brain dissections. Brains were dissected from the head capsule directly into RNAlater; RNA was extracted from individual brains and then pooled after extraction into caste-specific pools; pooling after RNA extraction allowed for elimination of any samples with low quality RNA. Pooling individuals was generally necessary to ensure sufficient RNA for analyses given the

small amount of brain tissue available for many species, as well as accounting for individual variation to ensure expression differences are due to caste or species, and not dependent on colony or random differences between individuals. One exception to this was the *V. vulgaris* samples which were sequenced as individual brains and pooled bioinformatically after sequencing. Individual sample sizes per species are given in Supplementary Data 1.

Total RNA was extracted using the RNeasy Universal Plus Mini kit (Qiagen, #73404), according to the manufacturer's instructions, with an extra freeze-thawing step after homogenization to ensure complete lysis of tissue, as well as an additional elution step to increase RNA concentration. RNA yield was determined using a NanoDrop ND-8000 (Thermo Fisher Scientific). All samples showed A260/A280 values between 1.9 and 2.1. An Agilent 2100 Bioanalyser was used to determine RNA integrity. Samples of sufficient quality and concentration were pooled and sent for sequencing. Libraries were prepared using Illumina TruSeq RNA-seq sample prep kit at the University of Bristol Genomics Facility. Five samples were pooled per lane to give ~50 million reads per sample. Paired-end libraries were sequenced using an Illumina HiSeq 2000. Pooling of individuals into single representatives of caste are described in Supplementary Data 1.

### Preparation of de novo transcriptomes

Transcriptomes of *Agelaia, Angiopolybia, Metapolybia, Brachygastra, Polybia, Polistes* and *Mischocyttarus* were assembled using the following steps. First, reads were filtered for rRNA contaminants using tools from the BBTools (version:BBMap_38) software suite (https://jgi.doe.gov/data-and-tools/bbtools/). We then used Trimmomatic v0.39[74] to trim reads containing adapters and low-quality regions. Using these filtered RNA sequences, we could assemble a de novo transcriptome for each species (merging queen and worker samples) using Trinity v2.8[75]. We then filtered protein-coding genes to retain a single transcript (most expressed) for each gene, which we use for the rest of the analyses.

For *Vespula* and *Vespa*, reads from both queen and worker samples were assembled into de novo transcriptomes using a Nextflow pipeline (https://github.com/biocorecrg/transcriptome_assembly). This involved read adapter trimming with Skewer v0.2.2[76], *de-novo* transcriptome assembly with Trinity v2.8.4[75] and use of TransDecoder v5.5.0[75] to identify likely protein-coding transcripts, and retain all translated transcripts. These were further filtered to retain the largest open-read frame-containing transcript, which we listed as the major isoform of each protein. Trinity assembly statistics are shown in Supplementary Data 2.

### Identification of orthologs

To identify gene-level orthologs, we used Orthofinder v.2.2.7[38] with diamond blast v0.9.22.123[77], with multiple sequence alignment using Muscle v3.8.31[78] and tree inference using FastTree v2.1.10[79], for our nine species (Supplementary Fig. 1). The largest spliced isoform per gene (from Trinity) was designated as the representative sequence for each gene. In addition, for specific iterations, we allowed the existence of multiple gene isoforms belonging to the same species in a single orthogroup (potential gene duplications); using up to 3 gene isoforms, only keeping the gene most highly expressed. This strategy allowed us to test a greater number of orthogroups, where a single duplication in one species would otherwise remove an entire orthogroup from the study. The drawback is that some gene isoforms are removed from the study that could have been informative. Similarly, we were able to include orthogroups that had missing gene information (NAs) for up to two species, filling in these missing genes with read counts of 10 for both queen and worker, thereby allowing us to test orthogroups where some species data are missing. Missing gene orthologs in a single species could be due to a true absence (likely gene loss) in a species, or due to assembly weaknesses, where the use of non-genome guided

RNA-Seq assemblies from just brain tissue, could lead to missing orthologs. Scripts to perform these steps are included in our Github repository (https://github.com/Sumner-lab/Multispecies_paper_ML/tree/master/Scripts_to_improve_orthogroups).

Orthofinder (with optional settings listed at the start of this section) was also used to create the phylogenetic tree in Supplementary Fig. 1, using the nine species in this manuscript with the additional of *Apis mellifera* (GCF_000002195.4), *Nomia melanderi* (GCA_003710045.1), *Dinoponera quadriceps* (GCA_001313825.1) and *Solenopsis invicta* (GCF_016802725.1), using only single copy orthogroups.

### Gene expression and differential expression

We calculated abundances of transcripts within queen and worker samples using "align_and_estimate_abundance.pl" within Trinity, using estimation method RSEM v1.3.1[80], "trinity_mode" and bowtie2[81] aligner. We then used edgeR v3.26.5[82] (R version 3.6.0) to compare gene expression between queens and workers. Given that we were comparing a single sequencing pool of several individuals per caste, we used a hard-coded dispersion of 0.1 and the robust parameter set to TRUE to account for sample size ($n = 1$). Raw $P$ values for each gene were corrected for multiple testing using a false discovery rate (FDR) cut-off value of 0.05. We did not take advantage of genome data (where available), as only two of the species had published genomes at the time of analysis; using transcriptome-only analyses makes the analysis more consistent across species. Trinity assemblies and RSEM counts are available on Gene Expression Omnibus (GEO; GSE159973 [https://www.ncbi.nlm.nih.gov/geo/query/acc.cgi?acc=GSE159973]).

To compare differential expression across the nine species (Fig. 3), we used the orthology list based on three splice isoforms represented as the most expressed and using only orthogroups with a single entry in all nine species after the merging of three isoforms. To ensure that the RNA-Seq reads for each species were of good quality we performed FASTQC analysis, this provides phred scores, duplicate read and GC content (see Supplementary Data 1), which showed that there were no major concerns. We also wrote a simple contamination pipeline using Nextflow[76] (https://github.com/chriswyatt1/Protein_blast_and_taxonomy), which blasts (diamond blastp) a single protein per gene (longest), against the whole of the 'nr' (non-redundant) database from the National Centre for Biotechnology Information (NCBI; downloaded:1.12.2021), to give a score for each taxonomic level for the top hit of each protein. In this way, we could confirm that the majority of assembly transcripts belong to Hymenoptera (Supplementary Data 1; Tabs 'Top_Genus'/'Top_Phylum'). To test for significant overlap of differentially expressed genes (Fig. 3) across the nine species by chance, we performed a permutation test ($n = 1000$), taking all orthologous genes from Supplementary Data 3, and randomly selecting genes of the same number as were differentially expressed in each individual species, then checking how many genes overlapped between the 9 total species. We could then use a Fisher's exact test (two-sided) for our comparisons of observed against expected (code available: https://github.com/Sumner-lab/Multispecies_paper_ML/tree/Version_2.0/Script_DEG_permutation).

### Building a normalised gene expression matrix

Focused on our set of one-to-one orthologs (merging 3 or less isoforms per species), we began by computing log transformed TPMs (transcripts per million reads) for each gene in each sample from the raw counts, followed by quantile normalisation. Next, we normalised for species, using an approach that is comparable to calculating a species-specific z-score for each sample. Specifically, we transformed the expression scores calculated above by subtracting the species mean and dividing by the species mean for each sample within a species. This calculation has two important effects. Firstly, subtracting the species mean from each sample within a species centres the mean expression of each species on zero, making the units of expression more

comparable across species. Secondly, dividing by the species mean from each sample standardises the expression scores, producing a measure that is independent of the units of measurement, so that the magnitude of difference between queens and workers in each sample is no longer important. The transformed expression score thus allows us to focus on the relative expression in queens versus workers across species. Finally, we removed orthogroups where the counts per million were below 10 (with some deviations in specific iteration; listed with run plots) in both Queen and Worker samples of each species, to remove lowly expressed genes that may contribute noise to subsequent analyses. We then performed principal component analysis (PCA) in R on the raw TPM values and those with the normalisation steps outlines above (see: https://github.com/Sumner-lab/Multispecies_paper_ML/blob/master/Scripts_to_run_svm/template_scripts/getExpressionAllOrthofinder.noNameReplace.R for full R code of this section).

### Machine learning (support vector machines)

Support vector machine (SVM) was used to classify caste across the species. We note that SVMs are more commonly used on larger datasets[83], but is known to have reasonable success with small datasets[84] where there are thousands of features (genes) can be used to classify smaller numbers of samples. In brief, this analysis involved taking species-scaled, logged and normalised gene expression data from the nine species (with one replicate for the queen and worker sample), filtering the lowly expressed genes, splitting into training and testing datasets (8 training, 1 testing), creating an SVM classifier that best separates caste in the training species (finding the best gamma and C tuning variables) and testing the one species left out, resulting in a caste classification. The code to run these steps is shown on github (https://github.com/Sumner-lab/Multispecies_paper_ML) and is executed in perl using subsidiary R scripts. The full details of these steps are outlined below, and in addition, a one liner Nextflow pipeline is also available (https://doi.org/10.5281/zenodo.7521827), using the same base code (detailed later).

The first step is to normalise the expression matrix, given the SVM classifier expects fully normalised and scaled data. This has been outlined in the previous section, and involves orthologous TPMs being logged, quantile and species-scaled; as well as filtering of lowly expressed genes. Next, data were split into test and train sets, so each species was addressed in turn, with queen and worker samples registered based on sample names (e.g. caste <- grepl("Worker ", colnames(data.train))), and converted into a binary format (where 1 is queen and 0 is worker). Then, a SVM classifier could be tuned using the train data (8 of 9 species), using the R package e1071[85], with optional grid search ranges of gamma = 10^(−8:−3) and cost (C) = 2^(1:10). These settings allow the classifier to explore the best gamma and cost settings to use in the model.

Given the majority of genes are not likely to contribute to caste differentiation (and therefore not help the SVM classifier), we then performed feature selection, which is commonly used in machine learning to increase the power of the classifier[86]. For this we used linear regression of each gene on caste (lm(caste ~ expr, data)), using the training data only. This results in regression beta coefficients for each orthologous gene, which we could then rank by their statistical association with caste (Supplementary Data 4 using the absolute values of the regression coefficients). This enabled us to measure the classification estimation as we filtered out genes statistically un-associated with the reproductive division of labour (Fig. 4a).

SVMs use a variety of kernels, which are the underlying algorithms used to classify the data in a single plain, these include radial, polynomial, linear and sigmoidal. To explore which kernel was most appropriate, we ran our SVM pipeline using these kernels (Supplementary Fig. 2). Based on these predictions with the different kernels,

we decided to continue with the radial kernel for the rest of our analysis, as it has decent prediction accuracy once the dataset is filtered, and has progressive improvement across the filtering spectrum (Supplementary Fig. 2; top left).

To further support our predictions, we plotted learning curves using the training sets (with 8 species; one species removed each time) at each percentage reduction in the feature selection step and calculated the cross-fold validation error (cross = 6) (Supplementary Data 9). When measuring the error across the whole dataset, we recorded an error of ~0.9 (for the nine leave-one-out replicates), yet once feature selection had been applied, the error was reduced to less than 0.05 with ~20 % of the features remaining in the majority of species. This suggests that the model is accurate when using the feature selection filter.

To help with interpreting the SVM plots, classification estimates of 0.5 indicate the SVM could not tell the difference between the two castes (maximal uncertainty), a classification certainty of 0 would indicate maximal certainty of a worker and 1 a maximal certainty of being a queen.

Finally, we also made a Nextflow v21.10.4.5656[76] pipeline of the entire SVM procedure (https://doi.org/10.5281/zenodo.7521827), where it is possible to try the same commands used to create SVM predictions in this paper. Running Nextflow in the way described, will download the Trinity assemblies and RSEM count data, and run the SVM in a container (using either singularity/docker) and produce the progressively filtered SVM plots shown in Fig. 4.

### Rates of gene evolution - dNdS scores

Using the single-copy orthologous genes for the nine species, we aligned nucleotide sequences for each of the 1,971 orthogroups (single-copy allowing for 3 isoforms) that successfully went through PRANK (version v.151120[87]) (Supplementary Data 7, tab ST1). For each species and for each orthogroup, we then calculated the ratio of rates of non-synonymous substitutions (dN) and the rates of synonymous substitutions (dS) in branch models (Supplementary Data 7, tabs ST2-ST16, and Supplementary Fig. 5; null hypothesis model in PAML codeml (model = 0, NSSites = 0, tree file without foreground) and in an alternative hypothesis model (model = 2, NSSites = 2, tree file without some taxa as foreground)). We then estimated episodic events of positive evolution when dN/dS >1 on a given phylogenetic branch (foreground) compared to the other background branches, in several tests contrasting lineages and social complexity (ST17-ST26; null hypothesis model (omega fixed at 1, model = 2, NSSites = 2, tree file with one species branch as foreground) and in an alternative hypothesis model (omega not fixed, all other parameters set as the null model) (version v. 4.8[88])). We assessed log likelihoods using a chi-square test and reported the adjusted P values using R (version 3.6.3) and Benjamini and Hochberg adjustment[89].

### GO enrichment and BLAST

To perform gene ontology (GO) enrichment tests, we used the R package TopGO v2.42.0[90], using Bonferroni cut-off P values of < =0.05. To assign gene ontology terms to genes in our new species, we used our orthofinder homology table with annotations to *Drosophila melanogaster* (downloaded from Ensembl Biomart 1.10.2019). Within species, we calculated enrichment of each species' gene to a background of all the genes expressed above a mean of 1 TPM. When comparing across the orthogroups (OG), we used *Metapolybia* GO annotations (derived from homology to *Drosophila*), with a background of those genes that have a mean >1 TPM in all species orthologues. GO comparisons were similar using other species as a database of gene to GO terms.

Blast2GO v 1.4.4[91] with default settings was used to annotate *Metapolybia* genes.

**Reporting summary**

Further information on research design is available in the Nature Portfolio Reporting Summary linked to this article.

## Data availability

The raw sequencing (RNA-Seq) data generated in this study as well as normalised Trinity count data have been deposited in the NCBI GEO (Gene Expression Omnibus) database under accession code GSE159973. The Supplementary Data provide the details of RNA-Seq sample (Supplementary Data 1), orthology (Supplementary Data 2), differentially expressed genes (Supplementary Data 3), SVM results (Supplementary Data 4–6), DnDs analysis (Supplementary Data 7), morphometrics data (Supplementary Data 8, including all measurements made and morphometric statistics) and learning curve error statistics (Supplementary Data 9).

## Code availability

Custom Nextflow/perl/R code used to automate the support vector classification and feature selection is provided in a GitHub repository (https://doi.org/10.5281/zenodo.7521827) with instructions for use. Scripts to improve orthology are found on Github: https://github.com/Sumner-lab/Multispecies_paper_ML/tree/master/Scripts_to_improve_orthogroups. Code to automate blast hits is found here: https://zenodo.org/record/7521834.

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

## Acknowledgements

We would like to thank Laura Butters for her help with the morphometric analyses of *Brachygastra*, Robin Southon, Daniel Fabbro, Liam Crowley and Sam Morris for their assistance in the field, James Carpenter at the American Natural History Museum and Christopher K. Starr for confirming species identification, and the Bristol Genomics Facility for their assistance with library preparations and sequencing. This work was conducted under collection and export permits for Trinidad (Forestry Division Ministry of Agriculture: #001162) and Panama (Autoridad Nacional del Ambiente (ANAM) SE/A-55-13). It was funded by the Natural Environment Research Council (NE/M012913/2; NE/K011316/1) awarded to SS, and a NERC studentship and Smithsonian Tropical Research Institute pre-doctoral fellowship awarded to E.F.B.

## Author contributions

S.S. conceived the study and supervised the project; S.S., E.L., E.B. and B.T. collected the samples; D.T., E.B., B.T. and R.B. performed molecular lab work; D.T. & R.B. carried out the morphological work; E.F., M.B. & C.W. executed the bioinformatics pipelines, performed the statistical analyses; C.W. & S.S. drafted the manuscript, with input from all authors.

## Competing interests

The authors declare no competing interests.
