## [Peer Review File · Nature Communications]

Social complexity, life-history and lineage influence the molecular basis of castes in vespid waspsReviewers' Comments:

Reviewer #1:

Remarks to the Author:

This manuscript is the product of considerable and sophisticated work. The authors state to have found evidence of a shared genetic toolkit across nine wasp species representing different levels of social complexity, but also evidence for shifts in the gene networks regulating social behaviour and rates of gene evolution that are influenced by innovations in both social complexity and life-history. I agree, and this manuscript will be of interest to a general reader. However, I am troubled by their outliers, *Brachygastra* and *Agelaia*, which clustered in opposite directions in the first analysis. According to the authors, these two taxa exhibit the most social complexity among their Polistinae. Why should this be so? This leads me to infer that something is being missed by the authors. I should like to see the authors offer more than "These two outlier species may not share the same caste-specific patterns as the other species, but we cannot rule out data and/or sampling anomalies."

There are a number of minor errors that are probably inevitable with a manuscript consisting of so many elements: Sumner (2014) appears twice in the literature citations; "were did" on line 492; there are 19 genera of Epiponini, not 21; there isn't a tribe Vespini.

Reviewer #2:

Remarks to the Author:

General Points

The submitted study explores eusocial evolution using transcriptomics and machine learning. An understudied group (wasps other than polistes) are studied and relatively novel methods based on machine learning are used to good effect to show that a conserved toolkit of genes underlies caste differences, but is not the only explanatory model necessary. The authors highlight the role of lineage and life history as well as social complexity in defining the nature of genomic evolution. Overall, I think this is a great study and worthy of publication in this journal. My comments relate to only two points. First, the citation record is weak in some places, and second, some of the conclusions drawn may depend on the choice of tissue (brain) used in the study. Some discussion of the possibility that the results may be tissue specific is necessary.

Minor Points

Line 57: this should probably be, "...evolution eusociality" as sociality is too broad

Line 84: should cite Mary Jane West Eberhards pioneering studies in which she initiates the study of ground plans. Amdam and Hunt should be cited as well.

Line 90: shouldn't the references 13-16 go in order of discovery? Jasper et al 2015 (MBE) should also be cited here

Line 98: the argument here is the explicit thesis of "Johnson and Linksvayer. "Deconstructing the superorganism: social physiology, groundplans, and sociogenomics." *The Quarterly Review of Biology* 85.1 (2010): 57-79.", so this should be cited.

Why brain transcriptomes? Most work in this field uses the brain, but this tissue may be the least amenable to study given its complexity. This is not a critique of the present study, but rather a suggestion to the authors to look at more tissues in future studies.

Lines 188-192: what is the justification for these cutoffs? 2 and 4 out of 9 do not seem to indicate

deep convergence. 2 is only 22%. How many orthologs play conserved roles at every cutoff (all, 8, 7, 6, etc)?

Lines 381: this almost certainly has to do with the use of brain transcriptomes. Its very possible that the patterns found in the brain would be flipped for other tissues. In fact, Jasper et al (2014) suggests this, as the brain is the least derived tissue in terms of novelty at the sequence level, at least in the honey bee. Thus, the argument that life history and lineage are important are probably true, but tissue specificity of the results cannot be neglected and should be discussed at some point in the paper.

Line 476: Kapheim et al 2015 should be cited and discussed as it also shows difference in genomic evolution with grade of eusocial complexity.

Lines 511-526: another interpretation is that this result is dependent on the tissue, brain, and might change when different tissues are analyzed

Reviewer #3:

Remarks to the Author:

This study uses comparative transcriptomics to test the hypothesis that the genes responsible for regulating caste behavior are similar across species that vary in social complexity. It is clear that the authors put a great deal of care into species selection to ensure they had a wide representation of social organization within one family of wasps. This is the most expansive comparison of social evolution in wasps from a transcriptomics perspective, which is no small feat given that there are so few publicly available wasp genome assemblies. This study thus provides an important point of comparisons for other comparative studies of social evolution that have focused on bees or ants. While this project thus holds an important place in the field, I have two major critiques of the paper that would need to be addressed prior to drawing conclusions based on the results presented here.

First, there are some methodological limitations that make it difficult to place too much confidence in the results. Most concerning is the details of the sample collection. It would be nice to have a sense of how the samples were collected and their sample sizes to get a sense of whether queen-worker behaviors/states/dynamics were likely to be captured in a similar way between species. For example, if workers in some species happened to be foraging and others happened to be sleeping or engaged in aggression at the time of collection, this could impact the degree to which gene expression patterns would represent queen/worker differences in a general sense. Similarly, if the colonies were at different points in the colony cycle, this could have a major effect on brain gene expression differences between castes. Moreover, it sounds from the Methods that queens and workers were identified based solely on ovary activation and sperm presence. This seems like an imprecise, if not error-prone, method given that some workers in some eusocial species can mate and lay eggs or there could be gynes with unactivated ovaries in the nest. Ultimately, whole brain gene expression profiles are a snapshot in time, and it is difficult to know how representative these are of the processes regulating caste differences more generally.

Providing sample sizes would also give the reader an idea of how much variance was likely to play a factor in the differential gene expression differences between species. Table S1 seems to indicate that several species had just one biological replicate of pooled samples for queens and workers. This seems counter to what is described in the Methods, which seems to indicate there are replicates of 2-4 pooled samples per caste/species, so some additional clarity could be helpful. This $n = 1$ seems to be confirmed later in the Methods (L763). If this is truly the case that each species was represented by a single pooled sample of Queens and a single pooled samples of Workers, it seems difficult to draw too many comparative conclusions from the analyses. Although I am not an expert on SVM methods, 1 replicate of 8 species (or fewer in the case of the follow-up analyses) seems like an extremely small

training set for machine learning. A quick google search suggests that a 100 or fewer datapoints is considered a small training set. This dataset does not seem to come close to this. Nonetheless, it seems there are some basic steps that can be taken to determine if the training set is large enough, such as plotting learning curves of training set size vs classification errors. Was this done?

One additional point on methods, or the approach more generally, is that so few aspects of each species' life history were taken into consideration. There are many ecological variables that could influence gene expression or rates of molecular evolution. A comprehensive way to address this would be to include several axes of ecological and social variation in the models, but at a minimum they should be tested or discussed as alternative explanations for the observed data. For example, additional aspects of biological variation are considered toward the end of the Results section when the same set of analyses are applied to colony founding method (swarm vs independent). However, this seems like an afterthought and still leaves plenty of other ecological differences (e.g., tropical vs temperate, type of nest, diet, etc) ignored. Fig. 1 does not provide any information about the rest of ecology of the species.

A second major limitation of the paper is the presentation of the results. In several places, it was difficult to determine what exactly had been done and what significance should be placed on the results being described. In many places, this could be easily fixed by (a) presenting a clear hypothesis and prediction prior to describing the result and (b) providing a few additional details about the methods that were used to generate the result. Some examples are below.

I had a hard time following the results of the SVM analysis, because some of the statistical terms used were unfamiliar to me. What does it mean to select features based on a linear regression from left to right? What does 0.6 likelihood mean? Is this like probability? So 60% of samples would likely be classified correctly? (This seems to be confirmed later in the Methods, but it would have been useful while reading the results.) Also, it seems like the method was used to see how well Queens could be classified. Was this also performed for Workers? I wonder if Workers would be more or less easy to classify.

I was more excited about the use of SVM to train a classifier on species with simple societies and see how well it predicts caste in more complex societies. Notwithstanding my questions about classification confidence above, it seems this test performed fairly well. It is intriguing that the reverse did not perform well, which is what you might expect given hypotheses presented in previous literature (e.g., Johnson Linksvayer 2010 Q Rev Biol). In the Discussion this result was described as "unexpected", but it seems to me the Johnson & Linksvayer paper suggests this is exactly what one would expect, given the degree of specialization and emergent properties that emerge in the "superorganismal" species. However, I noted that few statistical results were provided in the text when describing this result and looking at Fig. 5b the scores do not look that bad! It might help bolster the conclusion if additional details were provided in the text.

I also had some difficulty connecting the results of the dN/dS analysis to the results based on gene expression. I think this section could benefit from some clear hypotheses and predictions. This would help the reader understand how the dN/dS results are meant to be interpreted. In some places dN/dS rates ≥ 1 are taken to indicate positive selection and in other places they are more accurately described as indicators of more rapid evolution, which could of course include neutral evolution. Some additional description of the methods in the main text might help to justify these inferences.

The sentence "Some orthogroups had experienced significant positive selection (i.e., on the given foreground branch, null model rejected, chi-square test $p < 0.05$, dNdS ≥ 1 ; a total of eleven orthogroups)" was especially confusing to this reviewer. Does this mean there were a total of 11 orthogroups with dN/dS > 1 for all 9 iterations of an individual species on the foreground branch? Or this is the number that was common to all 9? Some additional detail could help to clarify, particularly if placed within a hypothesis-prediction framework. I found the sentences following that describing the

individual orthogroups to be difficult to interpret, partially because it was not clear what these orthogroups really signify and so it was difficult to become too invested in learning their (putative and inferred) functions.

The conclusion of this whole section is that “the molecular processes regulating caste differentiation vary depending on the level of social complexity”. However, I am unclear as to how the dN/dS rates of single copy orthologs pertain to the regulation of caste differentiation. How do dN/dS rates map onto other features of each species ecology? Is there justification for interpreting rates of evolution solely through the lens of caste differentiation? Generally, I think this section could have much higher impact and clarity if some clear hypotheses and predictions were presented.

Overall, this study represents a lot of important work and the genomic resources alone will be highly valuable for the field. A comprehensive comparative study of wasp transcriptomics is highly valuable in a field that has been dominated by ant and bee research. That said, there are some major limitations that will need to be addressed before this reviewer feels confident in drawing too many conclusions from the results.

Response to Reviewers

Thank you for your time revising our manuscript. This has been extremely helpful
and we hope we have adequately answered your concerns.

Specifically, we have rewritten the introduction and results sections to improve the
flow of the paper (including the addition of hypotheses, as suggested by reviewer 3).
Also, we have taken care to improve the methods specifically and conducted new
analyses to show that the data are reliable, yet acknowledge the limitations of our
experimental design and the conclusions we can take from them.

Finally, we highlight the addition of a Nextflow workflow (not in the original
manuscript) to run the SVM analysis from this paper (Github:
https://github.com/chriswyatt1/Trinity_to_SVM/). This workflow automates the
downloading of the Trinity transcriptomes and RSEM count files from NCBI, and
running of the main SVM scripts to get predictions of caste. All required tools are
containerized within the pipeline, for simplicity. This allows other researchers to try
out this technique with altered settings/filters with the same input data from the
paper. We feel this vastly improves the reproducibility and interest of the paper to the
wider community.

In the word version of the article, you can find links to each comment written
(Rev1,2,3; Comment 1.1,1.2 etc.), which can help find specific lines where the
manuscript has changed. If using the PDF version, we have added line numbers to
direct you to the specific line.

**REVIEWER COMMENTS**

Reviewer #1 (Remarks to the Author):

This manuscript is the product of considerable and sophisticated work. The authors
state to have found evidence of a shared genetic toolkit across nine wasp species
representing different levels of social complexity, but also evidence for shifts in the
gene networks regulating social behaviour and rates of gene evolution that are
influenced by innovations in both social complexity and life-history. I agree, and this
manuscript will be of interest to a general reader. However, I am troubled by their
outliers, *Brachygastra* and *Agelaia*, which clustered in opposite directions in the first
analysis. According to the authors, these two taxa exhibit the most social complexity
among their *Polistinae*. Why should this be so? This leads me to infer that something
is being missed by the authors. I should like to see the authors offer more than
"These two outlier species may not share the same caste-specific patterns as the
other species, but we cannot rule out data and/or sampling
anomalies."

Thank you for your review, we are pleased you highlighted the relevance and effort
gone into our publication. We really appreciate the time you have spent reviewing
our paper.

**Rev 1 Comment 1:** With regards to the two outlier species (*Brachygastra* and
*Agelaia*), we agree that these deserve more discussion of possible biological and
technical explanations. We now provide information on how we ruled out possible
technical explanations for these outliers. Below is a brief summary of what we've
added.

First, we determined whether there could be any concerns with respect to the quality
of the sequence data. We conducted a FASTQC analysis, measuring quality
statistics for each of our input reads (this has been added to Supp Table 1; we can
also provide the full html output if preferred). These results show that in general the
samples are of good quality, with phred scores mostly above 30 (error 1/1000; or
99.9% accurate), towards 40 (error 1/10000; or 99.99%) (Supp. Table 1; tab: "per
base quality stats"). Two species (*Metapolybia* and *Mischocyttarus*) had quality
scores that fell below 30 towards the end of the sequence; however, this is not of
substantial concern as phread scores just below 30 are typically still considered
decent quality ([https://www.illumina.com/documents/products/technotes/technote_Q-
Scores.pdf](https://www.illumina.com/documents/products/technotes/technote_Q-Scores.pdf)). Aside from this, there was no reason to doubt the quality of the two
outlier species (*Brachygastra* and *Agelaia*) based on phred scores.

A second quality control analysis we conducted was to examine GC content. The
FASTQC analysis showed that in general GC content of the raw RNA-Seq data was
normally distributed, although *Brachygastra*, *Metapolybia* and *Mischocyttarus* deviate
a little from normal (with additional peaks at >55% GC content; in both queen and
worker samples). We explored whether this could explain why *Brachygastra* is an
outlier in our analysis of caste-biased gene expression. The GC content of the toolkit
genes found in Fig. 4a in the SVM analysis (which are orthologous across all the nine
species) was 35.6% in *Brachygastra* for the 259 genes. This is similar to the mean of
all species (~36% to 42% GC, number found in Supp. Table 1 tab: General stats [%
GC]), suggesting that the genes used in this analysis are unlikely to be affected by
any anomalies in GC content in the outlier species. See Supplementary Table S1.

Third, we explored whether contamination could explain the outliers. Without a
genome for each species it is difficult to prove that each Trinity transcript came from
wasps with confidence, and that it does not contain contamination from other
species. To explore this, we blasted (diamond blastp; nr protein database
downloaded 1.12.2021; default settings) the expected proteins recovered from the
Trinity assemblies to check if the top hit for each gene came from an expected wasp-
like sequence (using a custom built Nextflow pipeline:

https://github.com/chriswyatt1/Protein_blast_and_taxonomy). This script creates
charts of the combined tags from kingdom to subspecies level for each protein,

which we included in Supp. Table 1 (Genus and phylum levels only; tab:
“Top_phylum” and “Top_genus”). This analysis shows that the majority of assembled
translated genes appear to be from arthropods; moreover, by genus, we mainly find
wasps with the correct genus level mappings expected, with little contamination from
other species. Where sequences matched closer to non-arthropod sequences
(Supp. Table 1; tab: “Top_phylum”), the majority come from proteobacterial genes
(the most common brain bacteria), fungi (inc. *Wolbachia*) and occasional
mammalian (inc. Human) sequences. However, these small levels of contamination
do not affect our analysis because we focus on orthologous genes across all nine
species – any contaminants in one or a few species would by definition not be
included. This is now referred to in the main text (**Rev 1 Comment 1.1 [line 337],**
**Rev 1 Comment 1.4 [line 193]**) and the discussion (Rev 1 Comment 1.4 [line 510])

A final technical explanation could be in the collection/sampling methods. This is
unlikely as all species were collected in a similar way: *Agelaia* was even collected in
the same place and time as other species (e.g. *Metapolybia*). However, as we note
in the methods, we were only able to sample one colony of *Agelaia*, and so we
cannot rule out the possibility that something was unusual about this colony (e.g. lost
its queens; recently predated; stage in colony cycle). This is included in the collection
methods (comment: **Rev 1 Comment 1.2, line 706**), and is now mentioned as a
possible explanation in the discussion (**Rev 1 Comment 1.3, line 593**).

In summary, we can see no technical reasons to exclude any samples due to any
sampling or processing anomalies, and no technical explanations for why *Agelaia*
and *Brachygastra* appear to be outliers in our analysis. These are explained in the
results section (**Rev 1 Comment 1.4, line 510-517**), and we now have included a full
table outlining all the quality scores (within Supplementary Table 1).

In the absence of a technical explanation, let’s now consider possible biological
explanations for why these two species are outliers. The reviewer rightly notes that
they cluster differently from the other species; this suggests that these two species
do not use the “toolkit” genes for caste differentiation in the same way as the other
wasp species. What is biologically different about these two species that could
explain this? There is no clear phylogenetic explanation as they are epiponines,
along with *Angiopolybia* and *Metapolybia*; it is unlikely that any environmental
differences can be an explanation as these two species occur in similar
tropical/subtropical conditions to all the other Polistines used in our study and in the
case of *Agelaia* they were sampled concurrently (in the same general location and
time period) as several of the other species. The main difference is that *Agelaia* and
*Brachygastra* represent intermediate levels of social complexity, as explained in
Figure 1: their societies are not as simple as *Polistes*, *Mischocyttarus*, *Angiopolybia*
or *Metapolybia*, but they are less socially complex than the superorganismal
vespines. Our analysis suggests that the same conserved social toolkit was likely
present in the common ancestor, and is also present in the highly complex societies.

but may be less important in species that are intermediate in complexity. One
interpretation of this is that a different set of genes regulates caste differentiation in
these intermediate forms of social complexity. If these species are indeed proxies for
the different stages of the major transition to superorganismality (as is generally
thought), then it is possible that *Agelaia* and *Brachygastra* represent a stage in the
major evolution transition that involves a possible upheaval of molecular processes
regulating castes, where pre-imaginal caste commitment is partially apparent.
Alternatively, these forms of social complexity may be unconnected with any
evolutionary patterns, and are an example of where a conserved molecular toolkit for
castes is not upheld. In this case, it is very interesting (albeit unexplained) that these
societies with intermediate levels of social complexity are regulated using different
processes to other types of societies. We are unable to address these ideas further
without wide taxonomic sampling. This is outside the scope of this study, but we
have included a new section in the discussion to address this.

Currently, the most parsimonious explanation is that there is a biological explanation
for these samples being a little different to the other 7 species but that further
sampling would be required before we can be sure how to interpret these outliers.
Overall, though, our main conclusion stands: we have found a robust set of genes
that are caste-biased across a wide diversity of social wasps. This remains the first
multi-species transcriptomic analysis of its kind in wasps. We hope it will stimulate
further genomic analyses of this interesting group

**Rev 1 Comment 2:** There are a number of minor errors that are probably inevitable
with a manuscript consisting of so many elements: Sumner (2014) appears twice in
the literature citations; "were did" on line 492;

Thanks, these issues have been fixed. Line 535

**Rev 1 Comment 3:** there are 19 genera of Epiponini, not 21;

We have updated the numbers of swarm founding wasps to reflect Menezes 2020
(<https://doi.org/10.1098/rspb.2020.0480>): (Epiponini include over 19 genera with at
least 246 species), and included the reference. Line 658

**Rev 1 Comment 4:** there isn't a tribe Vespini.

We have removed tribes from Figure 1, as they are not referred to anywhere else in
the MS and thus carry little relevant information. Line 1267

Reviewer #2 (Remarks to the Author):

General Points

The submitted study explores eusocial evolution using transcriptomics and machine
learning. An understudied group (wasps other than polistes) are studied and
relatively novel methods based on machine learning are used to good effect to show

that a conserved toolkit of genes underlies caste differences, but is not the only
explanatory model necessary. The authors highlight the role of lineage and life
history as well as social complexity in defining the nature of genomic evolution.
Overall, I think this is a great study and worthy of publication in this journal. My
comments relate to only two points. First, the citation record is weak in some places,
and second, some of the conclusions drawn may depend on the choice of tissue
(brain) used in the study. Some discussion of the possibility that the results may be
tissue specific is necessary.

Thank you for your kind words about our publication. We really appreciate the time
you have spent reviewing our paper.

**Rev 2 Comment 1:** We have added the helpful references you have listed in your
review. We now have 96 references; we hope now we have adequately covered the
literature within this manuscript.

**Rev 2 Comment 2:** The use of brain tissue. We agree that the choice of tissue
could have important implications for our conclusions. Historically, brain tissue has
been the favoured tissues for analyses of caste differentiation across all the social
insects as the brain is the epicentre of control for behaviour and because it is known
to be highly dynamic and responsive to social interactions and social behaviour (e.g.
reviewed in Traniellio & Robinson 2021; <https://doi.org/10.1146/annurev-neuro-092820-012959>). Our choice of brain tissue is because in these very simple wasp
societies, behaviour is the only difference between castes, so we expect the brain to
give us a more information about what defines caste. We have rewritten the
introduction section to explain why we chose brain in this study (**Rev 2 Comment**
**2.1, line 126**), and added a new section in the discussion to explain what patterns
we may expect in other wasp tissues (**Rev 2 Comment 2.2, Lines 509-604**), which
would be important to explore in future work.

Minor Points

**Rev 2 Comment 3:** Line 57: this should probably be, "...evolution eusociality" as
sociality is too broad

Done, line Line 57

**Rev 2 Comment 4:** Line 84: should cite Mary Jane West Eberhards pioneering
studies in which she initiates the study of ground plans. Amdam and Hunt should be
cited as well.

We now cite West Eberhard 1996 (line 81). We omit the diapause hypothesis of
Amdam and Hunt though, as this is a more specific mechanistic scenario by which
the decoupling might occur, and with already 90+ references, we prefer to limit the
reference list to those that are directly relevant to the MS. The West-Eberhard
reference is the original one, and correct to cite here. Thanks for flagging this to us.

Rev 2 Comment 5: Line 90: shouldn't the references 13-16 go in order of discovery? Jasper et al 2015 (MBE) should also be cited here
Apologies for this, we were using Mendeley, which seems to have a lot of issues. We have gone through all the references and made sure they are in order of date in the first instance, then by number of original reference in subsequent references (as per Nature Communications standards). Thanks for spotting this.

Rev 2 Comment 6: Line 98: the argument here is the explicit thesis of "Johnson and Linksvayer. "Deconstructing the superorganism: social physiology, groundplans, and sociogenomics." The Quarterly Review of Biology 85.1 (2010): 57-79.", so this should be cited.

These citations have been updated (reference 24). Lines 98,140,141,506,510,

Rev 2 Comment 2 (repeated): Why brain transcriptomes? Most work in this field uses the brain, but this tissue may be the least amenable to study given its complexity. This is not a critique of the present study, but rather a suggestion to the authors to look at more tissues in future studies.

See answer above. But, a further comment here too. We agree that a more extensive sampling of different tissues may lead to more genes that have caste-biased function. Recently, some studies have included analyses of other tissues; e.g. Kapheim et al 2020 studied expression in different life stages, abdomen and brain tissue of a facultatively eusocial bee; Jasper et al 2014 examined expression differences among different tissues of the honeybee workers. We agree with the reviewer that a deeper understanding of caste differentiation warrants more exploration of differential gene expression in a range of relevant tissues (including subsections of brains, like mushroom bodies (Tranellio & Robinson 2021). But this is out of scope for the current study. As stated above, though, we now add some text about this in the Introduction and Discussion (**Rev 2 Comment 2.1 [line 126]and 2.2 [lines 509-604]**).

Rev 2 Comment 7: Lines 188-192: what is the justification for these cutoffs? 2 and 4 out of 9 do not seem to indicate deep convergence. 2 is only 22%. How many orthologs play conserved roles at every cutoff (all, 8, 7, 6, etc)?

We now provide all the cutoffs to make this part clearer - we have added a histogram to Figure 3 (part A: **Rev 2 Comment 7, line 1322**) to show the numbers at each cut-off. This highlights the fact that we do not recover many differentially expressed genes across all nine species. Our use of 2+ and 4+ cut-offs were arbitrary. We chose 2+ (original plot 3B) for the Gene Ontology (GO), so that we included sufficient genes to run the analysis, and we showed the names and expression patterns of the 4+ genes (original plot 3C) due to space available in the plot (3+, would be 166 lines, too many to show clearly on a page, with minimal font size). We provide the full list

in Supplementary Table S3. To be consistent, we added a new GO plot (part B-
using DEGs in any direction; n=552; before it was in the same direction) and now
highlight which plots are coming from the cumulative values in the new plot 3A. The
terms are similar to the previous result, but now we can clearly show in the figure
where these gene sets come from

**Rev 2 Comment 8:** Lines 381: this almost certainly has to do with the use of brain
transcriptomes. Its very possible that the patterns found in the brain would be flipped
for other tissues. In fact, Jasper et al (2014) suggests this, as the brain is the least
derived tissue in terms of novelty at the sequence level, at least in the honey bee.
Thus, the argument that life history and lineage are important are probably true, but
tissue specificity of the results cannot be neglected and should be discussed at
some point in the paper.

The reviewer is referring to our finding that rates of evolution were higher in the
simpler societies than the more complex ones, and that this is counter to previous
studies on bees which found higher rates in the more complex social species. This
cannot be explained by the tissue choice, as the study we are comparing our data
with (Shell et al 2021) was also on brains (whole heads), as indeed are many other
studies on DEG in social insect castes. However, the choice of tissue is an important
one (raised earlier by this reviewer), and certainly brain tissue appears to have fewer
highly expressed differentially expressed genes compared to other tissues. As
explained above, we have now added some text to justify our choice and
acknowledging the possible limitations (see “Rev 2 Comment 2” changes in MS).

**Rev 2 Comment 9:** Line 476: Kapheim et al 2015 should be cited and discussed as
it also shows difference in genomic evolution with grade of eusocial complexity.

This has now been cited (**Rev 2 Comment 9, line 519**)

**Rev 2 Comment 10:** Lines 511-526: another interpretation is that this result is
dependent on the tissue, brain, and might change when different tissues are
analyzed.

We disagree: the study we are comparing our results with (Shell et al 2021) was
conducted on brains (heads) too. However, as explained above, we add a sentence
noting that analysing other tissues would also be useful (see Rev 2 Comment 2
changes in MS).

Reviewer #3 (Remarks to the Author):

This study uses comparative transcriptomics to test the hypothesis that the genes
responsible for regulating caste behavior are similar across species that vary in
social complexity. It is clear that the authors put a great deal of care into species

selection to ensure they had a wide representation of social organization within one
family of wasps. This is the most expansive comparison of social evolution in wasps
from a transcriptomics perspective, which is no small feat given that there are so few
publicly available wasp genome assemblies. This study thus provides an important
point of comparisons for other comparative studies of social evolution that have
focused on bees or ants. While this project thus holds an important place in the field,
I have two major critiques of the paper that would need to be addressed prior to
drawing conclusions based on the results presented here.

**We thank the reviewer for their time spent reviewing our manuscript, and appreciate**
**the kind comments acknowledging the amount of work and care we put in to this**
**manuscript, and its importance to the field.**

**Rev 3 Comment 1:** First, there are some methodological limitations that make it
difficult to place too much confidence in the results. Most concerning is the details of
the sample collection. It would be nice to have a sense of how the samples were
collected and their sample sizes to get a sense of whether queen-worker
behaviors/states/dynamics were likely to be captured in a similar way between
species. For example, if workers in some species happened to be foraging and
others happened to be sleeping or engaged in aggression at the time of collection,
this could impact the degree to which gene expression patterns would represent
queen/worker differences in a general sense.

Similarly, if the colonies were at different points in the colony cycle, this could have a
major effect on brain gene expression differences between castes. Moreover, it
sounds from the Methods that queens and workers were identified based solely on
ovary activation and sperm presence. This seems like an imprecise, if not error-
prone, method given that some workers in some eusocial species can mate and lay
eggs or there could be gynes with unactivated ovaries in the nest. Ultimately, whole
brain gene expression profiles are a snapshot in time, and it is difficult to know how
representative these are of the processes regulating caste differences more
generally.

**We had already provided quite some detail about the collection methods, sample**
**sizes (both number of colonies and number of individuals) and how we characterised**
**the castes for each species. Specifically:**

- **the first paragraph of the Methods section entitled ‘Sample collection’ (Rev 3**
**Comment 1.1, line 701) gives the number of colonies sampled for each**
**species; these are also listed in Suppl. table S1. In the same section we**
**explain how samples were pooled to generate caste-specific transcriptomes;**
**again, this includes sample sizes but also this is given in complete detail in**
**Suppl. Table S1.**

- **We also provided some extensive morphometrics datasets and analyses of**
354 **three species (*Brachygastra*, *Polybia* and *Metapolybia*) as explicit data on the**

355 level of caste allometry were lacking in the literature – these are found in
Supplementary Table S8 and methods given in the Method section entitled
‘Morphometrics’ (See Rev 3 Comment 1.2, lines 672-674).

- - We explicitly acknowledge the problem stage of colony cycle (**Rev 3**
**Comment 1.3, lines 701-703**) and with the exception of one species
(*Agelais*), multiple colonies were sampled (**Rev 3 Comment 1.4, lines 705-**
**706**).
- - We state how castes were defined in the Section entitled “Identification of
queens and workers” (**Rev 3 Comment 1.5, line 723**): “Inseminated females
with developed ovaries were scored as ‘queens’; non-inseminated females
with undeveloped ovaries were scored as workers”. The same information is
also given in the Main Text (**Rev 3 Comment 1.6, line 159**): ..”adults for the
two main social phenotypes – adult reproductives (defined as mated females
with developed ovaries, henceforth referred to as ‘queens’ for simplicity; see
Supplementary Table S1) and adult non-reproductives (defined as unmated
females with no ovarian development, henceforth referred to as ‘workers’; see
Methods).”

However, in light of the reviewer’s concerns, we now add a new Methods
section that stipulates explicitly how caste was defined (**Rev 3 Comment 1.5,**
**line 723**). We also add further text about the types of colonies used (all post-
emergence), detailing on how we ruled out intermediate wasps (some ovarian
development, but not mated), callows (**Rev 3 Comment 1.7, lines 727-738**).
In this way we can be certain that we didn’t select reproductive workers,
gynes or callows. It is worth noting here that it is impossible to collect queens
from any of the epiponines or vespines without opening the nest because of
the envelop that surrounds them. There is therefore no other way to identify
these castes, other than through ovarian dissections. Indeed, the entire
literature on these insects is based on assigning caste from dissections. It is
hard to see how else this could have been done. With over 20 years of
experience working with social wasps (including dissecting them and studying
caste behaviours), we are very confident in our assignment of caste. The only
possible anomaly is *Agelais*, where (as the reviewer points out) we only
sampled from one colony as they are hard to find. Indeed, this could explain
why they were somewhat of an outlier (see response to Reviewer 1); we
discuss this in the Discussion now (see text highlighted Rev 1 Comment 1).

**Rev 3 Comment 2:** Providing sample sizes would also give the reader an idea of
how much variance was likely to play a factor in the differential gene expression
differences between species. Table S1 seems to indicate that several species had
just one biological replicate of pooled samples for queens and workers. This seems
counter to what is described in the Methods, which seems to indicate there are
replicates of 2-4 pooled samples per caste/species, so some additional clarity could
be helpful. This n= 1 seems to be confirmed later in the Methods (L763). If this is

truly the case that each species was represented by a single pooled sample of
Queens and a single pooled samples of Workers, it seems difficult to draw too many
comparative conclusions from the analyses.

It is a difficult trade-off in these types of analyses when resources are limited:
replication within species versus between. For our question, it was important to
maximise power in making comparisons *across* species, rather than *within* species.
We have now included in the results (mentioning “individuals and pools” were used
[Rev 3 Comment 2.1, line 158] and that “Using these data we could construct *de*
*novo* brain transcriptomes and estimate gene expression for a pool of queens and a
pool of workers for each species.” [Rev 3 Comment 2.2, lines 163-165]). Due to the
size of some of our specimens, we needed to pool in order to get sufficient RNA for
sequencing (explained in Rev 3 Comment 2.3, line 170). Others could be
sequenced as individuals; e.g. *Vespula*. But in order to make the comparisons
across species, we pooled these individual brain RNA-Seq datasets computationally
(explained in Rev 3 Comment 2.4, lines 775-776). We have added a ReadMe tab to
Supp. table 1, to further explain. Pooling allows us to remove individual-level
variance in expression. Therefore, despite low levels of technical replication within
species, we have good levels of biological representation within species and across
species, allowing us to look for conserved patterns in a comparative context. We
acknowledge the need for higher levels of technical replication in future studies, in
the discussion.

**Reviewer 3 Comment 3:** Although I am not an expert on SVM methods, 1 replicate of
8 species (or fewer in the case of the follow-up analyses) seems like an extremely
small training set for machine learning. A quick google search suggests that a 100 or
fewer datapoints is considered a small training set. This dataset does not seem to
come close to this. Nonetheless, it seems there are some basic steps that can be
taken to determine if the training set is large enough, such as plotting learning curves
of training set size vs classification errors. Was this done?

SVMs are used routinely in biomedical research, where they have the benefit of
large sample sizes. The sample used for our SVMs is small compared to this field.
However, there are examples where machine learning techniques have been used
on small datasets with less than 8 training samples (including:
<https://www.ncbi.nlm.nih.gov/pmc/articles/PMC8296305/>) and similarly find that it
has been a useful to find patterns in their datasets compared to looking at
differentially expressed genes alone. There has also been a review of SVM
techniques with different datasets, showing that the technique is still relevant with
smaller datasets
(<https://www.tandfonline.com/doi/full/10.1080/24751839.2019.1660845>). We are
aware of our limitations here, but for our field, our application of machine learning is

novel, and we hope it will inspire others to use similar methods in the future. We
have added this reference to the main text (see Rev 3 Comment 3.1, lines 887-889).

We had removed the learning curves from the original manuscript, thinking the
predictions alone would be sufficient. In the light of the reviewer's comments, we
now show the learning curves and predicted error of the model (see Supplementary
Table S9 (with addition methods section **Rev 3 Comment 3.2, lines 926-929**)). Using 6
449 fold cross validation of eight species (repeated; with one species left out), that the
450 cross validation error using all the data was around 0.9 (showing that the predictions
were not correct, using all the data). Yet, when you use feature selection the error
reduced to ~ 0.05 when using $\sim 20\%$ of the most informative genes (after feature
selection), in the nine replicates with a species left out. Having error scores lower
than 0.05 show that the svm model is accurate and consistent across different input
datasets. We hope this helps show that the data are consistent in their predictive
power over the different feature selection filters.

We propose that it is sufficient to make our conclusions by training on 8 species and
testing on the ninth. We argue that even with our low number of samples, given we
are using gene expression data with thousands of features, that we can make
accurate predictions. For example, many biomedical papers (e.g.
<https://www.sciencedirect.com/science/article/abs/pii/S0169260714003228?via%3Di>
hub) have less than 40 features per sample (instead of thousands in our dataset).
This is highlighted in the result shown in Figure 4a, where after feature selection, we
get the expected predictions of caste in the majority of species, suggesting that there
is sufficient signal in the training sets to make a prediction. However, these
classification estimates rarely get above 0.6, suggesting that these predictions are
not clear cut, almost certainly due to the low sample number in the analysis.

Ultimately, SVMs are powerful at detecting small changes across large number of
samples, yet they can still perform well enough even with small datasets. We add a
comment in the methods and discussion acknowledging the concerns about using this
approach on smaller datasets (**Rev 3 Comment 3.3, lines 499-501**).

**Rev 3 Comment 4:** additional point on methods, or the approach more generally, is
that so few aspects of each species' life history were taken into consideration. There
are many ecological variables that could influence gene expression or rates of
molecular evolution. A comprehensive way to address this would be to include
several axes of ecological and social variation in the models, but at a minimum they
should be tested or discussed as alternative explanations for the observed data. For
example, additional aspects of biological variation are considered toward the end of
the Results section when the same set of analyses are applied to colony founding
method (swarm vs independent). However, this seems like an afterthought and still

leaves plenty of other ecological differences (e.g., tropical vs temperate, type of nest,
diet, etc) ignored. Fig. 1 does not provide any information about the rest of ecology
of the species.

This comment surprised us, as we *did* consider some of the effects of ecological and
demographic variables, whereas few studies on caste genomics do. We devote an
entire analysis to this very question. Fig. 1 includes the key ecological variables that
are thought to be important in wasp social behaviour and ecology. All social wasps
are generalist predators, with a preference for lepidopterans and dipterans.

Regarding the type of nest, the main difference is that the independent founding
Polistines build nests without an envelope, whilst all the others have envelopes. It is
not possible to examine the contribution of this nest type variable to caste-biased
gene expression because it is confounded by both phylogeny and level of social
complexity. Regarding tropical/temperate: all the Polistines (which happen to also
be those with simpler societies) sampled are tropical and all the Vespines (which
happen to also be the only truly superorganismal species) sampled are temperate;
so, again, these variables are confounded. However, the reviewer raises some valid
points and so we have made the following changes.

- - We have revised Figure 1 to reflect these other life history traits; specifically,
we now add more of the life history traits to Figure 1 and expand the legend.
- - To try to explore the possible effects of life-history a little further, we
conducted a FAMD plot (Factor analysis of mixed data; see below), showing
that queen and worker differences were the greatest explaining variable in the
dataset, with the other life-history variables not contributing as much to these
two top dimensions. We do not currently choose to include this in our revision,
as we feel that the text deals with this. However, if the reviewer or editor
would like us to, this can be added to the methods or supplementary
document.

Review Figure 1. Dimension reduction plot using FAMD (Factor analysis of mixed data). Plot the dimension reduction to two components using all the orthologous genes in the nine species (with species normalization step), then applying categorical data to the samples, including reproductive (Queen) or not (Worker), plus nest-type, envelope type, nest-found method, colony size and allometry state. The lines emanating from the centre show in which plane the different ecological variables have greatest effect, with the black triangles showing the outlier points of Queen and Worker identity.

**Rev 3 Comment 5:** A second major limitation of the paper is the presentation of the
results. In several places, it was difficult to determine what exactly had been done
and what significance should be placed on the results being described. In many
places, this could be easily fixed by (a) presenting a clear hypothesis and prediction
prior to describing the result and (b) providing a few additional details about the
methods that were used to generate the result. Some examples are below.

I had a hard time following the results of the SVM analysis, because some of the
statistical terms used were unfamiliar to me.

535 A) We have clarified the hypotheses and predictions in the Introduction and
536 throughout the results section. (**Rev 3 Comment 5.1, lines 132-134,140,313,**
**etc.**)

B) We have entirely overhauled the text of the methods and SVM results section.
Some of these details are in the comments below. (see **Rev 3 Comment 5.2,**
**lines 885-934**)

What does it mean to select features based on a linear regression from left to right?

This meant that from left (or 99% in x axis) to right (1%), the percentage of the
remaining features chosen using linear regression, for which we plotted the predicted
classification scores. Linear regression is picking out genes that are informative with
regard to caste, and the predictions therefore improve if we focus on the subset of
genes different between the castes in the training dataset. This of course will have
no impact on the test dataset, which is not used in the linear regression. We have
rewritten this methods and results section to better explain this, and in figure
legends. **Rev 3 Comment 5.3 (lines 1344-1346)**

What does 0.6 likelihood mean? Is this like probability? So 60% of samples would
likely be classified correctly? (This seems to be confirmed later in the Methods, but it
would have been useful while reading the results.)

We have changed this term in the manuscript now, only using the term “classification
estimate” (e.g. **Rev 3 Comment 5.4, line 259**), instead of likelihood, which is stated,
as we cannot take statistical likelihood directly from SVM models (although this can
be accomplished using Platt scaling;
<https://home.cs.colorado.edu/~mozer/Teaching/syllabi/6622/papers/Platt1999.pdf>),
so “classification estimate” is more appropriate and hopefully not misleading as
‘likelihood’ is in our case. We have changed this term throughout the paper and
expanded both the results and the methods section to fully explain this classification
score. We have also added a section in the SVM results, to clarify what the 0 and 1

scores mean in terms of worker and queen, which was lacking (**Rev 3 Comment**
**5.5, lines 248-252**).

Also, it seems like the method was used to see how well Queens could be classified.
Was this also performed for Workers? I wonder if Workers would be more or less
easy to classify.

**Workers have exactly the opposite score of the queen for each prediction (e.g 0.9 in**
**queen will be -0,9 in worker, for the same species). This is due to our pooling of**
**replicates to one sample for each caste, so when we scale from -1 to 1, queens and**
**workers fit on the opposite scale.**

**Rev 3 Comment 6:** I was more excited about the use of SVM to train a classifier on
species with simple societies and see how well it predicts caste in more complex
societies. Notwithstanding my questions about classification confidence above, it
seems this test performed fairly well. It is intriguing that the reverse did not perform
well, which is what you might expect given hypotheses presented in previous
literature (e.g., Johnson Linksvayer 2010 Q Rev Biol). In the Discussion this result
was described as “unexpected”, but it seems to me the Johnson & Linksvayer paper
suggests this is exactly what one would expect, given the degree of specialization
and emergent properties that emerge in the “superorganismal” species.

**We have now integrated the predictions of Johnson & Linksvayer 2010 more**
**explicitly into the text in the introduction and discussion, and reworded to reflect the**
**reviewer’s point. (Rev 3 Comment 6.1, line 149).**

However, I noted that few statistical results were provided in the text when
describing this result and looking at Fig. 5b the scores do not look that bad! It might
help bolster the conclusion if additional details were provided in the text.

**Yes, we agree that we should have included some of the main numbers in the text**
**(now included; Rev 3 Comment 6.2, lines 350-351). Some of the predictions were**
**indeed not so bad in Figure 5b (inc. Mischocyttarus), but used a much smaller list of**
**the genes after filter selection to obtain this. This suggests that with much greater**
**levels of filtering you can still obtain semi-decent classifications. We have now**
**rewritten this section to highlight the fact that the simple society trained SVM used a**
**much larger gene set (~800), with high levels of correct classification compared to**
**the complex society trained SVM (~350), this is perhaps more informative about the**
**predictive ability of the two sets. (Rev 3 Comment 6.2, lines 350-351).**

**We have also improved the legend in Figure 5, to help with interpreting the plots.**
**(Rev 3 Comment 6.3, line 1368).**

**Rev 3 Comment 7:** I also had some difficulty connecting the results of the dN/dS
analysis to the results based on gene expression. I think this section could benefit
from **some clear hypotheses and predictions**. This would help the reader
understand how the dN/dS results are meant to be interpreted. In some places
dN/dS rates ≥ 1 are taken to indicate positive selection and in other places they are
more accurately described as indicators of more rapid evolution, which could of
course include neutral evolution. Some additional description of the methods in the
main text might help to justify these inferences.

The sentence “Some orthogroups had experienced significant positive selection (i.e.,
on the given foreground branch, null model rejected, chi-square test $p < 0.05$, dNdS
$\Rightarrow 1$; a total of eleven orthogroups)” was especially confusing to this reviewer. Does
this mean there were a total of 11 orthogroups with dN/dS > 1 for all 9 iterations of
an individual species on the foreground branch? Or this is the number that was
common to all 9? Some additional detail could help to clarify, particularly if placed
within a hypothesis-prediction framework. I found the sentences following that
describing the individual orthogroups to be difficult to interpret, partially because it
was not clear what these orthogroups really signify and so it was difficult to become
too invested in learning their (putative and inferred) functions.

**We have rewritten this section to make the predictions explicit (lines 374-), we now**
**explicitly mention that dN/dS ratios strictly above 1 (i.e. not equal to 1, representing**
**neutral evolution, line 854) represent episodic events of positive selection. As a brief**
**summary, we describe the overall ratios of dN/dS in simpler societies compared to**
**more complex societies, testing for significant differences (Wilcoxon test). We found**
**that the simpler societies had experienced a higher rate of evolution than the**
**complex societies. This average is however masking out the direction of selection**
**pressures for individual loci. Thus we also listed the eleven orthogroups with dNdS**
**ratio above 1, with associated BLAST result. For each result, we added the**
**hypothesis driven prediction so that the reader can follow our analyses.**

**Rev 3 Comment 8** The conclusion of this whole section is that “the molecular
processes regulating caste differentiation vary depending on the level of social
complexity”. However, I am unclear as to how the dN/dS rates of single copy
orthologs pertain to the regulation of caste differentiation. How do dN/dS rates map
onto other features of each species ecology? Is there justification for interpreting
rates of evolution solely through the lens of caste differentiation? Generally, I think
this section could have much higher impact and clarity if some clear hypotheses and
predictions were presented.

**Rev 3 Comment 8.1:** Re: How do dN/dS rates map onto other features of each
species ecology? **This is addressed under the new header of Hypothesis 5 in the**

Intro and results. As explained above, the only life-history trait that varies in a way
that is independent of social complexity is nest founding. We did indeed report
differences in rates of gene evolution with respect to nest founding; this is detailed in
the Results. We now add a comment in the discussion that the choice of species for
analysis in future studies may afford further testing of the relationship between other
life-history traits and gene evolution. See line 536 onward.

**Rev 3 Comment 8.2:** Re: Is there justification for interpreting rates of evolution solely
through the lens of caste differentiation? **Yes.** We expanded this section to address
the reviewer's valid points. It is widely thought that in hymenopterans where the
queen reproduces and the workers do not reproduce (caste differentiation), the
molecular regulators of the worker phenotype are free to evolve more rapidly than
those regulating the queen phenotype. Based on this, we predicted that worker-
biased genes to experience rapid rate of evolution (see Privman *et al.* 2018;
<https://doi.org/10.1111/mec.14767>). Thus, we examine the rates of molecular
evolution among, specifically, the caste biased genes (dN/dS and DEG) to examine
the effects of selection pressures between reproductives and non-reproductives
across lineages. The results are also highlighted in the main text in response to Rev
3 Comment 7 (line 396 onward).

Overall, this study represents a lot of important work and the genomic resources
alone will be highly valuable for the field. A comprehensive comparative study of
wasp transcriptomics is highly valuable in a field that has been dominated by ant and
bee research. That said, there are some major limitations that will need to be
addressed before this reviewer feels confident in drawing too many conclusions from
the results.

Reviewers' Comments:

Reviewer #2:

Remarks to the Author:

I am satisfied with the revisions.

Reviewer #3:

Remarks to the Author:

The primary aim of the study is to determine whether there is a conserved genetic toolkit for sociality. The authors report that this toolkit does exist, but that other lineage-specific and life-history-specific factors matter too. This is not a terribly surprising conclusion, but it is important to describe it in this understudied group of social insects.

One major question I have is about what it takes to determine if there is a conserved toolkit or not. Are the number of shared caste-biased genes across species greater than would be expected by chance? For example, if you randomly selected the same number of genes as were considered differentially expressed across species, how many times would there be 57 common to 4 or more species (Fig 3b)? It seems this kind of permutation test (or something similar) would have to be done before labeling the results as evidence for a conserved genetic toolkit. What if there were only 20, 10, or 5 genes differentially expressed across 4 species? Would this still be evidence for a shared toolkit? It seems some minimum number of genes would show up on multiple gene lists by chance, right? What is the null hypothesis? Alternatively (or perhaps concomitantly), you could look at the proportion of differentially expressed genes within a species and ask if more than expected by chance are shared with one or more other species.

I also wondered about alternative explanations. In testing Hypothesis 2, could the reason for lower classification accuracy when going from complex to simple societies be related to phylogenetic history? The four species included in the complex society group are more distantly related to one another (and may also have more diverse ecological niche space (e.g. both temperate and tropical vs all tropical)?) than those included in the simple society groups. Could this also potentially explain the results of testing Hypothesis 5? (The swarm founders are more closely related than the independent founders.) I guess this is tested in Hypothesis 4, but perhaps the two sets of results could be more integrated to strengthen the story?

I appreciate the authors' justification and rationale for the lack of biological replicates within species and the use of SVM with small sample sizes. While I can appreciate the limitations in place, this does not change the fact that conclusions drawn from such small sample sizes are potentially unreliable.

L165 – I would make it abundantly clear that you used a *single* biological replicate for queen and worker for each species.

Rebuttal

REVIEWER COMMENTS

Reviewer #2 (Remarks to the Author):

I am satisfied with the revisions.

We are delighted that this Reviewer is happy with our revision. We thank them for their time, once again.

Reviewer #3 (Remarks to the Author):

Thank you for your time helping us improve this manuscript.

The primary aim of the study is to determine whether there is a conserved genetic toolkit for sociality. The authors report that this toolkit does exist, but that other lineage-specific and life-history-specific factors matter too. This is not a terribly surprising conclusion, but it is important to describe it in this understudied group of social insects.

Comment 3.0

We should have stressed greater at the beginning that in general, wasps with simple societies are known to have very small numbers of differentially expressed genes between castes (a result also supported by our results). This meant that we were not certain that across so many species of wasps, that we would find a social toolkit. I have added this information in the introduction, as it was lacking, to highlight the fact that finding a social toolkit was not inevitable. We have added at line 123: “, *known to have few caste-biased differential genes, potentially suggesting that a toolkit may not be present across wasps*⁵.”

One major question I have is about what it takes to determine if there is a conserved toolkit of not. Are the number of shared caste-biased genes across species greater than would be expected by chance? For example, if you randomly selected the same number of genes as were considered differentially expressed across species, how many times would there be 57 common to 4 or more species (Fig 3b)? It seems this kind of permutation test (or something similar) would have to be done before labeling the results as evidence for a conserved genetic toolkit. What if there were only 20, 10, or 5 genes differentially expressed across 4 species? Would this still be evidence for a shared toolkit? It seems some minimum number of genes would show up on multiple gene lists by chance, right? What is the null hypothesis? Alternatively (or perhaps concomitantly), you could look at the proportion of differentially expressed genes within a species and ask if more than expected by chance are shared with one or more other species.

Comment 3.1

Thanks for raising this; it's important. We have now added the following analyses. First, we now show hypergeometric tests for significant pair-wise overlap of these differentially expressed genes across the nine species (see Fig 1 below; this could

be added as a Supp. figure if the editor/reviewer prefers). Second, we performed a permutation test (as suggested, with 1000 permutations), taking all orthologous genes from Supplementary Table S3, and randomly selecting genes of the same number as were differentially expressed in each individual species, then checking how many genes overlapped between the 9 total species. With this, we could calculate a Fisher's exact test (two-sided) for our observed against expected. For example, in our study we observe 57 orthologs differentially expressed in 4 or more species, and calculated an average of 0.055 genes should be expected in 4 species (after 1000 permutations) by chance. We now show the expected, observed and Fisher's exact test in Figure 3, with a summary table below (Table 1); we have also added the method to the Methods section (L873, with the code on GitHub: https://github.com/Sumner-lab/Multispecies_paper_ML) and changed the figure legend on L1345.

Number of DEG species	1	2	3	4	5	6	7	8	9
Average	2825.015	89.632	2.497	0.055	0.002	0	0	0	0
Variance a	358.6954 7	85.854430 43	2.5705615 62	0.0580330 33	0.0019979 98	0	0	0	0
Observed	2204	552	166	57	20	5	2	1	0
Fisher Exact test (two-sided)	1.3635E-10	1.03517E-74	4.64596E-46	1.22026E-17	1.85036E-06	0.062393 43	0.0623934 28	1	1

Table 1. Table of permutations with the total number of single (1), double (2) etc. species DEG genes found (top line), and the predicted average number of orthogroups in each of the groups found (either just in one species(1), or in two (2), etc.). The third line shows the variance of all 1000 permutations. Finally, we list the observed numbers for each category, and their two-sided Fisher exact test p values, comparing the observed and expected values.

Figure 1. Hypergeometric test of significant overlap between differentially expressed genes between the nine species. Each square shows Bonferroni corrected P-values for Hypergeometric tests. Dark green highlights pairs with low P-values, scaled to white, where we have P-values near 1.

I also wondered about alternative explanations. In testing Hypothesis 2, could the reason for lower classification accuracy when going from complex to simple societies

be related to phylogenetic history? The four species included in the complex society group are more distantly related to one another (and may also have more diverse ecological niche space (e.g. both temperate and tropical vs all tropical)?) than those included in the simple society groups. Could this also potentially explain the results of testing Hypothesis 5? (The swarm founders are more closely related than the independent founders.) I guess this is tested in Hypothesis 4, but perhaps the two sets of results could be more integrated to strengthen the story?

Comment 3.2

It is true that the four complex species represent a more diverse group, so this factor should be fully explored. Given there are only two vespids, we couldn't run SVM predictions based purely on these two vespids vs the other polistines as a training with 2 samples is not possible (mentioned in L380). However, we have tried repeating some of the SVMs using slightly different training sets to assess the effect of phylogeny of the predictions we report in the paper. First, we trained our model on four Polistines and testing on the fifth (Metapolybia, Mischocyttarus, Polistes, Polybia and Angiopolybia; removing the weakly supporting species Agelaia), to see if training within its own clade would improve the classifications of each test. This does not improve the classification in four out of the five Polistine species; Metapolybia was the only species with improved classifications. This suggests that our findings do not seem to be affected greatly by distance within the wasp family tree. However, we agree that it is important to acknowledge (and we do stress this at line 389 and 607 onward), that these analyses would benefit from additional replicate species of both simple and complex species in future studies to ensure that the findings are consistent. But, as a first attempt (and with the caveats made explicitly, as we already do in the discussion) we feel this is an important result and worth reporting here. The reviewer's point that Hyp 4 also provides some support for the fact that effects of phylogeny are less than caste. But integrating the two analyses together, we feel, would make the hypotheses less clearcut to understand. However, in the light of this we have switched round the hypotheses 3 (DnDs) with those of 4 (sub-family) and 5 (life history), so that the subfamily section follows on directly from Hyp 2.

Finally, this new data is updated in Supp. Table 6 (SVM of subsections, and its README), a new Supp. Figure 3 (Sub-family tests only), separate from Supp. Figure 4 (Life history SVMs), with Supp. Figure 5 being that of the DnDs analysis. Legends have been updated and references to these figures in the main text.

I appreciate the authors' justification and rationale for the lack of biological replicates within species and the use of SVM with small sample sizes. While I can appreciate the limitations in place, this does not change the fact that conclusions drawn from such small sample sizes are potentially unreliable.

Comment 3.3

The level of technical replication within species is small, we agree. However, the biological replication contributing to these samples is typical (or exceeds) that of most studies of this kind (mostly between 3 to 11 individuals in a pool). Prior to our

study, studies on caste differentiation transcriptomics in wasps was limited to a few species from a single genus – Polistes, representing a single level of simple social complexity. Our study thus takes a big step forward in analysing data across nine species from two different subfamilies (there are only three subfamilies of social wasps), and nine different genera.

Nonetheless, the reviewer's concerns are valid – greater replication is always desirable; we had acknowledged and discussed this in the existing manuscript. E.g. line 347 (“future work should confirm this by expanding the repertoire of species”)/ and in the discussion, line 485: (“Wider taxonomic sampling is required to determine the extent to which the role of these additional processes is driven by the evolution of superorganismality”) and line 491: (“highlighting the importance of wider taxonomic consideration and replication in the quest to understand the molecular basis of sociality and the nature of the major transition”). See also response to Comment 3.2.

Comment 3.4

L165 – I would make it abundantly clear that you used a *single* biological replicate for queen and worker for each species.

L165, now makes this clear:

It is not correct that we used a single biological replicate: we pooled many biological replicates for each caste for each species into a single *technical* replicate.

“These were sequenced as individual pools to generate a worker sample and a queen sample (both made up of between 3 and 11 biological replicates per caste pool). “